# Multiomic single cell sequencing identifies stemlike nature of mixed phenotype acute leukemia

Cheryl A. C. Peretz[1,2,11], Vanessa E. Kennedy[3,11], Anushka Walia[3], Cyrille L. Delley[4], Andrew Koh[3], Elaine Tran[3], Iain C. Clark [5], Corey E. Hayford[6], Chris D'Amato[6], Yi Xue[6], Kristina M. Fontanez[6], Aaron A. May-Zhang [6], Trinity Smithers [6], Yigal Agam[6], Qian Wang[7,8], Hai-ping Dai [7,8], Ritu Roy [2], Aaron C. Logan [3], Alexander E. Perl[9], Adam Abate [4], Adam Olshen[2,10] & Catherine C. Smith [2,3] ✉

Despite recent work linking mixed phenotype acute leukemia (MPAL) to certain genetic lesions, specific driver mutations remain undefined for a significant proportion of patients and no genetic subtype is predictive of clinical outcomes. Moreover, therapeutic strategy for MPAL remains unclear, and prognosis is overall poor. We performed multiomic single cell profiling of 14 newly diagnosed adult MPAL patients to characterize the inter- and intra-tumoral transcriptional, immunophenotypic, and genetic landscapes of MPAL. We show that neither genetic profile nor transcriptome reliably correlate with specific MPAL immunophenotypes. Despite this, we find that MPAL blasts express a shared stem cell-like transcriptional profile indicative of high differentiation potential. Patients with the highest differentiation potential demonstrate inferior survival in our dataset. A gene set score, MPAL95, derived from genes highly enriched in the most stem-like MPAL cells, is applicable to bulk RNA sequencing data and is predictive of survival in an independent patient cohort, suggesting a potential strategy for clinical risk stratification.

Survival of patients with mixed phenotype acute leukemia (MPAL) is poor and inferior to that of the more common acute lymphoid and myeloid leukemias (ALL and AML)[1]. MPAL is characterized by leukemic blasts co-expressing both lymphoid and myeloid cell-surface markers or with co-existing populations of myeloid and lymphoid blasts. The diagnostic definition of MPAL remains unrefined. While both ALL and AML are defined by genetic drivers, the 2022 World Health Organization (WHO)[2] and International Consensus Classification[3] guidelines continue to define MPAL by immunophenotype with only a subset with associated genetic abnormalities (BCR::ABL1 fusion, KMT2A, ZNF384, and BCL11B rearrangements). Of note, some of these genetic aberrations are unique to pediatric patients[4,5], leaving the drivers of adult MPAL even less clear than its pediatric counterpart. Further, a large proportion of MPAL remains unassociated with these defining genetic abnormalities.

[1]Division of Hematology and Oncology, Department of Pediatrics, University of California San Francisco, San Francisco, CA, USA. [2]Helen Diller Family Comprehensive Cancer Center, University of California San Francisco, San Francisco, CA, USA. [3]Division of Hematology and Oncology, Department of Medicine, University of California San Francisco, San Francisco, CA, USA. [4]Department of Bioengineering and Therapeutic Sciences, University of California San Francisco, San Francisco, CA, USA. [5]Department of Bioengineering, University of California Berkeley, Berkeley, CA, USA. [6]Fluent Biosciences Inc., Watertown, MA, USA. [7]National Clinical Research Center for Hematologic Diseases, Jiangsu Institute of Hematology, The First Affiliated Hospital of Soochow University, Suzhou, People's Republic of China. [8]Institute of Blood and Marrow Transplantation, Collaborative Innovation Center of Hematology, Soochow University, Suzhou, People's Republic of China. [9]Department of Medicine, Division of Hematology-Oncology, Perelman School of Medicine at the University of Pennsylvania, Philadelphia, PA, USA. [10]Department of Epidemiology and Biostatistics, University of California San Francisco, San Francisco, CA, USA. [11]These authors contributed equally: Cheryl A. C. Peretz, Vanessa E. Kennedy. ✉e-mail: catherine.smith@ucsf.edu

Genomic alterations in MPAL are not unique and include mutations recurrently mutated in ALL or AML[6]. The biologic connection between immunophenotype and genotype in MPAL remains unknown. Importantly, neither the immunophenotype nor the genotype of MPAL correlate clearly with overall survival (OS) or treatment response, suggesting a more complete biologic understanding of MPAL is required to guide disease definition and risk stratification[2,7].

Due to the relative rarity and heterogeneous nature of MPAL, optimal therapeutic strategies remain uncertain. Emerging data suggest that sub-classification of MPAL may be needed to facilitate therapeutic decision making[8]. However, the full immunophenotypic, genetic, and transcriptomic profiles that may determine risk stratification of this complex disease have not been elucidated. Until recently, the technology to simultaneously determine immunophenotypic, genetic, and transcriptomic heterogeneity in MPAL has not existed. MPAL, with its definitionally "mixed" immunophenotype, is uniquely poised to benefit from multiomic single cell (SC) sequencing analysis, which can quantify the relationship between these biologic factors on a single cell level to better understand the biologic origin of MPAL and potential drivers of prognosis.

Here, we use multiomic SC profiling on newly diagnosed MPAL samples to characterize immunophenotypic, genetic, and transcriptional landscapes of adult MPAL. We identify MPAL as a stem-like leukemia with a shared gene expression signature. We further describe a transcriptional metric, derived from MPAL blasts with greatest differentiation potential, that is predictive of patient survival. These results broaden our understanding of MPAL biology and suggest a path toward risk stratification for a disease in which no risk stratification currently exists.

## Results

### The transcriptional landscape of MPAL
To characterize the genetic, transcriptional, and immunophenotypic landscape of MPAL, we analyzed samples from 14 patients with newly diagnosed MPAL using two SC technologies in parallel: CITE-seq (SC RNA plus protein sequencing)[9] and DAb-seq (SC DNA plus protein sequencing)[10–12] (Fig. 1a). Patient characteristics are in Supplementary Data 1. By clinical immunophenotyping via flow cytometry, our cohort included 10 patients with B/myeloid, 3 patients with T/myeloid, and 1 patient with B and T/myeloid MPAL.

A total of 72,131 individual cells from 12 patients were analyzed by CITE-seq (median 6010 cells/sample; range 1173–10,275) (Supplementary Data 2). Two additional patients had insufficient cells for CITE-seq analysis and were only profiled using DAb-seq. For CITE-seq analysis, we used a particle-templated instant partitions sequencing (PIP-seq) approach to perform SC indexing of transcriptomes and epitomes sequencing (CITE-seq) analysis with a panel of 19 barcoded antibodies (Supplementary Data 3)[9]. Across all patients, SC transcriptional data were integrated, clustered by transcription, and annotated (Fig. 1b). Notably, all 12 patients, regardless of MPAL immunophenotypic subtype, contributed to the cluster annotated as leukemia, and the common leukemia cluster contained single cells from diagnostic samples derived from both bone marrow and peripheral blood (Supplementary Fig. 1a, b). Each of the 12 patients contributed 4.5%–10.4% (median 8.8%) of the cells in the common leukemia cluster after normalization for number of cells isolated per patient. Furthermore, immunophenotypic subtype was not the primary predictor of transcriptional variation in correspondence analysis (Supplementary Fig. 2). Relative to non-leukemic cells and clusters, the leukemia cluster demonstrated a unique transcriptional signature, despite its heterogeneity (Fig. 1c; Supplementary Fig. 3; Supplementary Data 4, 5).

### Transcription alone does not determine immunophenotype
We next examined how gene expression was associated with immunophenotype in our integrated cohort. Across all cells and all patients,

through unsupervised clustering of immunophenotypic markers, we identified 13 immunophenotypically defined subpopulations. For many of these subpopulations, the cell type as identified by transcription closely associated with the expected immunophenotype (Supplementary Fig. 4). For example, transcriptionally defined normal T cells were composed of 87.2% CD3+/CD5+ cells, while transcriptionally defined normal B cells were 94.2% CD19+/CD22+ cells (Fig. 1d).

To contrast, across all patients, the transcriptionally defined "leukemia" cells were comprised of cells from heterogeneous immunophenotypic subpopulations, with the greatest contributions from cells with stem or myeloid markers, including CD34+/CD13+ cells (12.89% of leukemia population), CD34+/CD117+ cells (12.86%), CD33+/CD64+ cells (11.60%), and CD34+/CD33+/CD117+ cells (11.20%). Cells with lymphoid markers were also present in the transcriptionally defined leukemia cells, but in smaller proportions, including CD19+/CD22+/CD30+ cells (5.96%) CD19+/CD22+/CD45+ cells (5.49%), CD3+/CD5+/CD7+ cells (4.45%), and CD3+/CD4+/CD5+ cells (0.4%) (Fig. 1d). Importantly, within the integrated leukemia population, transcriptionally defined subpopulations did not cluster by immunophenotype (Fig. 1d). Similarly, when all leukemic cells were analyzed as immunophenotypically defined subpopulations, while there were some differences in gene expression, many subpopulations had markedly similar expression patterns (Fig. 1e). This reflects that many individual single cells and cell population had similar gene expression, despite having heterogenous immunophenotypes. There is no clear shared gene expression profile by immunophenotypic subtype.

On the individual patient level, the association between transcription and immunophenotype was heterogeneous, closely associating in 4/12 patients (33%) and not associating in 8/12 (66%). In some patients, immunophenotype was closely associated with a distinct transcriptional signature. For example, in Patient 11, immunophenotype-based clustering revealed distinct CD34+ and CD33+ populations (Fig. 1f). In addition to having distinct immunophenotypes, these two populations also had distinct gene expression profiles, with the CD33+ population demonstrating markedly higher expression of major histocompatibility complex-encoding genes relative to the CD34+ population (Fig. 1g). In other patients, however, immunophenotype and transcriptional profile were not closely associated. For example, in Patient 2, immunophenotype-based clustering also revealed distinct CD34+ and CD33+ subpopulations, but these two immunophenotypically distinct subpopulations did not have distinct transcriptional profiles (Fig. 1h, i).

### MPAL cells upregulate stem-like pathways and are distinct from genetically defined MPAL subsets
To further define the common transcriptional signature of MPAL, we performed unbiased single-cell gene set enrichment analysis (GSEA) on transcriptionally annotated leukemia cells systematically across all patients using all molecular signature database (MSigDB) hallmark and C2 gene sets (Fig. 2a)[13,14]. Single-cell GSEA demonstrated enrichment for gene sets associated with stem cells. Out of all gene sets, the greatest enrichment was demonstrated for a gene signature first described in CD133+ stem cells derived from human cord blood (normalized enrichment score [NES] 2.92, $q$ value 0.0); genes associated with embryonic stem cells were also highly enriched (NES 2.41) (Fig. 2b; Supplementary Data 6)[15–17]. Decreased enrichment was demonstrated in gene signatures associated with immune or inflammatory pathways, including natural killer cell cytotoxicity, complement activation, and interferon-gamma signaling (Supplementary Fig. 5).

We conducted a targeted assessment for the enrichment of known gene sets derived from multiple immature or lineage-ambiguous leukemias, including: early T-cell progenitor (ETP) ALL[18], KMT2A-rearranged B-cell ALL[19], early pro-B BCR-ABL + B-ALL[20], hematopoietic stem cell (HSC)-like AML[21], the acute myeloid leukemia stem

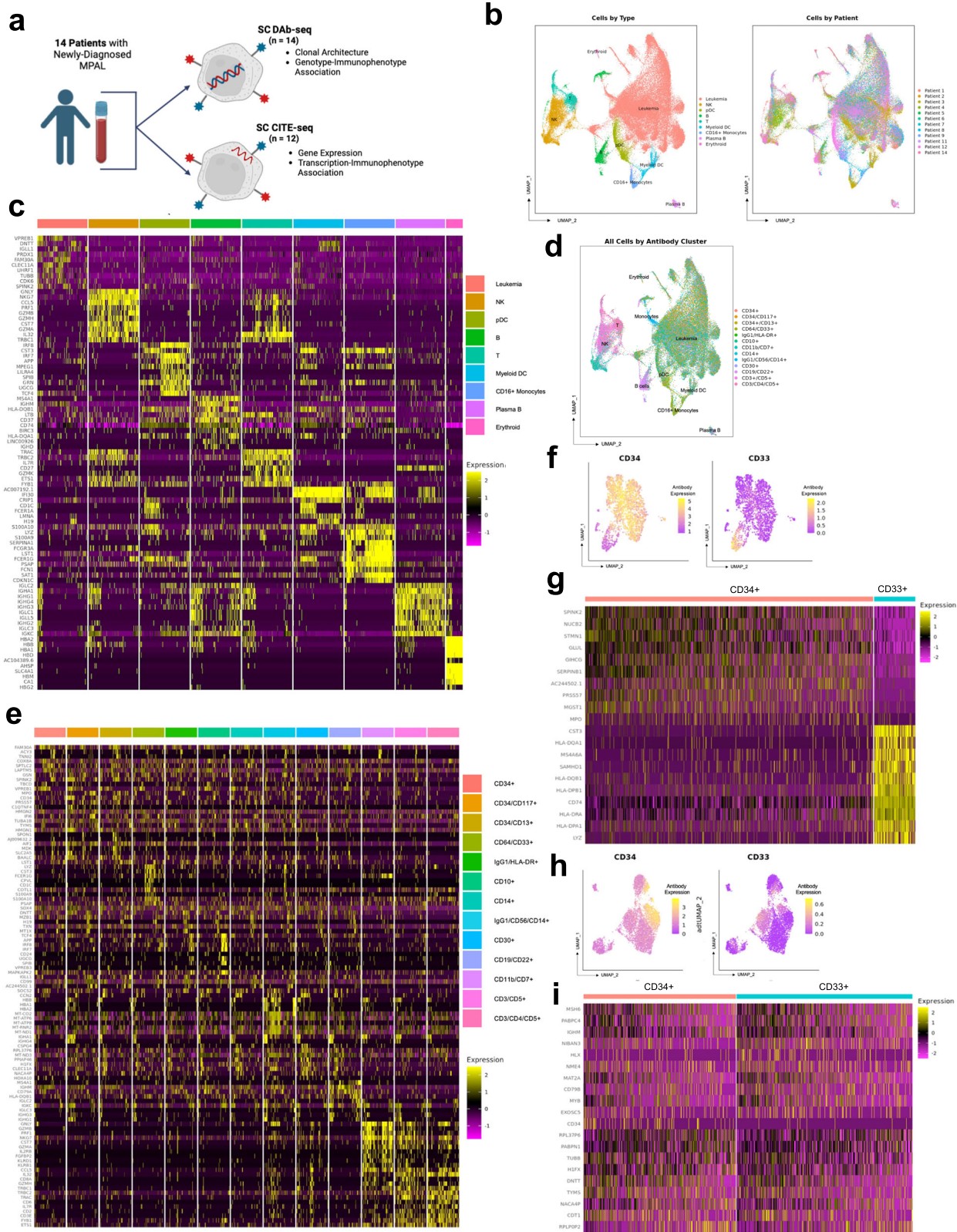

cell (LSC)-47[22], and B-ALL with subsequent monocytic lineage switch[23]. We also assessed gene sets derived from more differentiated acute leukemias, including granulocyte–monocyte progenitor-like AML[21], myeloid-like AML[24], *NUTM1*-rearranged ALL[19], and signatures for *BCR-ABL* + B-ALL spanning later B-cell differentiation[20].

Of these, only signatures associated with HSC-like AML[21], and LSC-47[22] were both significantly enriched (NES 2.15, *q* value 0.003; NES

2.07, *q* value 0.024), supporting MPAL as a stem-like leukemia (Fig. 2c; Supplementary Data 6, 7).

While many MPAL patients do not have characteristic genetic features, a subset of MPAL is associated with *BCL11B* and *ZNF384* rearrangements. More common in children, these rearrangements were not identified in our adult cohort (Supplementary Data 1), and gene sets associated with these rearrangements, including *TCF3-*

**Fig. 1 | MPAL is comprised of a common transcriptomic signature and heterogenous transcription-immunophenotypic associations. a** Schematic depicting sample workflow. Created with BioRender.com released under a Creative Commons Attribution-NonCommercial-NoDerivs 4.0 International license (https://creativecommons.org/licenses/by-nc-nd/4.0/deed.en). **b** RNA-derived UMAP from comprehensive SC CITE-seq analysis of 71,579 cells from 12 patients. Cells are color-coded by cell lineage/type as determined by gene expression data (left) and by individual patient (right). Source Data are provided as a Source Data file. **c** Heatmap of scaled expression values for top 10 most upregulated genes for each transcriptionally defined cell type as identified in (**b**). Source Data are provided as a Source Data file. **d** RNA-derived UMAP from (**b**). Cells are annotated based on transcriptionally defined cell populations, clustered by the expression of cell-surface immunophenotypic protein expression into 13 immunophenotype-defined clusters, and then color-coded based on cluster. Source Data are provided as a Source Data file. **e** Heatmap of scaled expression values for top 10 most upregulated genes in each of the 13 immunophenotypic subpopulations from (**d**). Source Data are provided as a Source Data file. **f** RNA-derived UMAP from 2594 cells from Patient 11. Cells are color-coded based on expression of CD34 (left) and CD33 (right). Source Data are provided as a Source Data file. **g** Heatmap of scaled expression values for top 10 most upregulated genes for the CD34-positive cell population (left columns) and the CD33-positive cell population (right columns) from Patient 11. Source Data are provided as a Source Data file. **h** RNA-derived UMAP from 6100 cells from Patient 2. Cells are color-coded based on expression of CD34 (left) and CD33 (right). Source Data are provided as a Source Data file. **i** Heatmap of scaled expression values for top 10 most upregulated genes for the CD34-positive cell population (left columns) and the CD33-positive cell population (right columns) from Patient 2. Source Data are provided as a Source Data file.

*ZNF384* B-ALL, *ZNF384*-rearranged B-ALL or MPAL, *BCL11B*-expressing CD34+ cells, and *BCL11B*-expressing T-ALL cells were not significantly enriched in MPAL leukemic blasts (Supplementary Data 6; Fig. 2a)[5,25–27]. As *BCL11B* rearrangements are associated with BCL11B over-expression, we also evaluated BCL11B expression in our cohort. Consistent with the genetic features, *BCL11B* was expressed in a small minority of cells (406 cells, 0.76%) in the common leukemia cluster and in <3% of cells in any individual patient (Supplementary Fig. 6a, b). BCL11B-expressing cells did not overexpress the conserved MPAL gene signature relative to non-BCL11B-expressing cells and there was no difference in OS for patients as stratified by percent of BCL11B cells (Supplementary Fig. 6c, d). *KMT2A and BCR::ABL1* rearrangements are also recurrently associated with MPAL. While our cohort includes patients with these rearrangements (3 and 1 each with KMT2A rearrangement and BCR::ABL, respectively) (Supplementary Data 1), blasts from these patients exhibited the same shared MPAL signature. Notably, a *KMT2A*-rearranged gene set[19] was not enriched in MPAL blasts from the three KMT2A-rearranged patients (or in the cohort as a whole). Overall, these data suggest that despite heterogenous underlying genetics, MPAL blasts share a gene expression profile similar to HSCs and distinct from previously identified gene signatures derived from MPAL genetic subsets.

## MPAL cells upregulate RUNX1-regulated gene expression programs

A recent study integrating single-cell transcription and chromatin accessibility in five adult MPAL patients found that RUNX1 motifs were the most commonly shared accessible elements[28]. In our cohort, *RUNX1*-regulated programs were similarly enriched. Pathway enrichment analysis of the greatest differentially upregulated genes in the common MPAL cluster against the ChIP-x Enrichment Analysis (ChEA) and Encyclopedia of DNA Elements (ENCODE) transcription factor targets databases via the enrichr platform identified RUNX1 as the most significantly enriched (odds ratio [OR] 10.2, $p = 1.3e - 5$) (Fig. 2d)[29,30]. Similarly, in GSEA, Reactome transcriptional regulation by *RUNX1* and targets of *RUNX1* in monocytes were significantly enriched as well (NES 2.06, $q = 0.028$ and NES 2.02, $q = 0.049$, respectively) (Supplementary Fig. 7a), and RUNX1 gene expression was increased in the leukemic population relative to non-leukemic cells (Supplementary Fig. 7b).

Three patients in our cohort had pathogenic *RUNX1* mutations as identified by DAb-seq (Patients 4, 6, and 11). To assess whether *RUNX1*-regulation transcription was enriched independent of *RUNX1* mutations, although *RUNX1* mutations are typically loss of function, we repeated the above analyses in the nine patients without *RUNX1* mutations. In this subset analysis, GSEA demonstrated similar enrichment for *RUNX1* regulation (Reactome transcriptional regulation by *RUNX1*: NES 2.13, $q = <0.001$; targets of *RUNX1* in monocytes: NES 1.94, $q = 0.003$) (Supplementary Fig. 7c). Similarly, pathway enrichment analysis of the conserved MPAL signature of the subsetted cohort

again demonstrated significant enrichment for RUNX1 targets (OR = 11.9, $p = 1.69e - 6$) (Supplementary Fig. 7d). Taken together, this emphasizes the potential importance of RUNX1 as a leukemic driver in adult MPAL, with or without known *RUNX1* mutation or rearrangement. In addition to RUNX1, the most significantly upregulated transcription factor programs identified by ChIP-x and ENCODE analysis (Fig. 2d) included KLF4 (OR 8.7, $p = 2.34e - 4$), a Yamanaka factor and known regulator of pluripotency[31,32], as well as NELFE (OR 15.8, $p = 0.0014$), an RNA binding protein implicated in regulation of gene signatures associated with MYC, another well-known pluripotency factor[33,34]. The upregulation of gene programs driven by KLF4 and associated with MYC further supports that the transcriptional signature of MPAL is fundamentally stem-like.

## The common MPAL gene expression signature is upregulated in an independent cohort

We next assessed whether the gene expression signature identified in the common leukemia cluster of our cohort was similarly upregulated in a separate validation cohort. To do this, we analyzed SC RNAseq data from an independent, previously published cohort of five adult patients with MPAL. In contrast to our cohort, in which 9/12 patients had B/Myeloid disease, 4/5 patients in this independent cohort had T/Myeloid disease (4 T/Myeloid, 1 B/Myeloid)[28]. A total of 11,133 single cells were integrated, clustered by transcription, and annotated using methods identical to those used in analysis of our cohort (Supplementary Fig. 8a). Differentially upregulated genes identified in our common leukemia cluster were similarly upregulated in the common leukemia cluster of the independent cohort (Fig. 2e).

We then performed a GSEA on the annotated leukemia cells from the independent cohort using the MSigDB hallmark, C2, and select gene sets derived from other leukemias, as described above. GSEA on the annotated leukemia cells demonstrated striking upregulation of our MPAL gene expression signature (NES 2.91; $q = 0.000$) (Fig. 2f); out of all gene sets assessed, this demonstrated the greatest enrichment (Supplementary Fig. 8b). Similar to our patient cohort, the leukemia cells from the independent cohort also demonstrated significant enrichment of stem cell gene sets and gene sets associated with stem-like AML; gene sets associated with immature ALL, differentiated leukemia, *KMT2A, ZNF384,* and BCL11B-rearranged leukemias were not enriched (Supplementary Fig. 8b–d). Of note, like our cohort, this comparison cohort did not include characteristic *ZNF384* or *BCL11B* rearrangements. Unlike our cohort, which included only samples from newly diagnosed patients, this cohort included newly diagnosed patients as well as patients previously treated with both AML and ALL chemotherapy regimens[28].

## The common MPAL gene expression signature is not upregulated in normal hematopoietic stem cells

To distinguish how stem-like MPAL blasts are transcriptionally distinct from normal HSCs, we performed SC RNAseq on a bone marrow

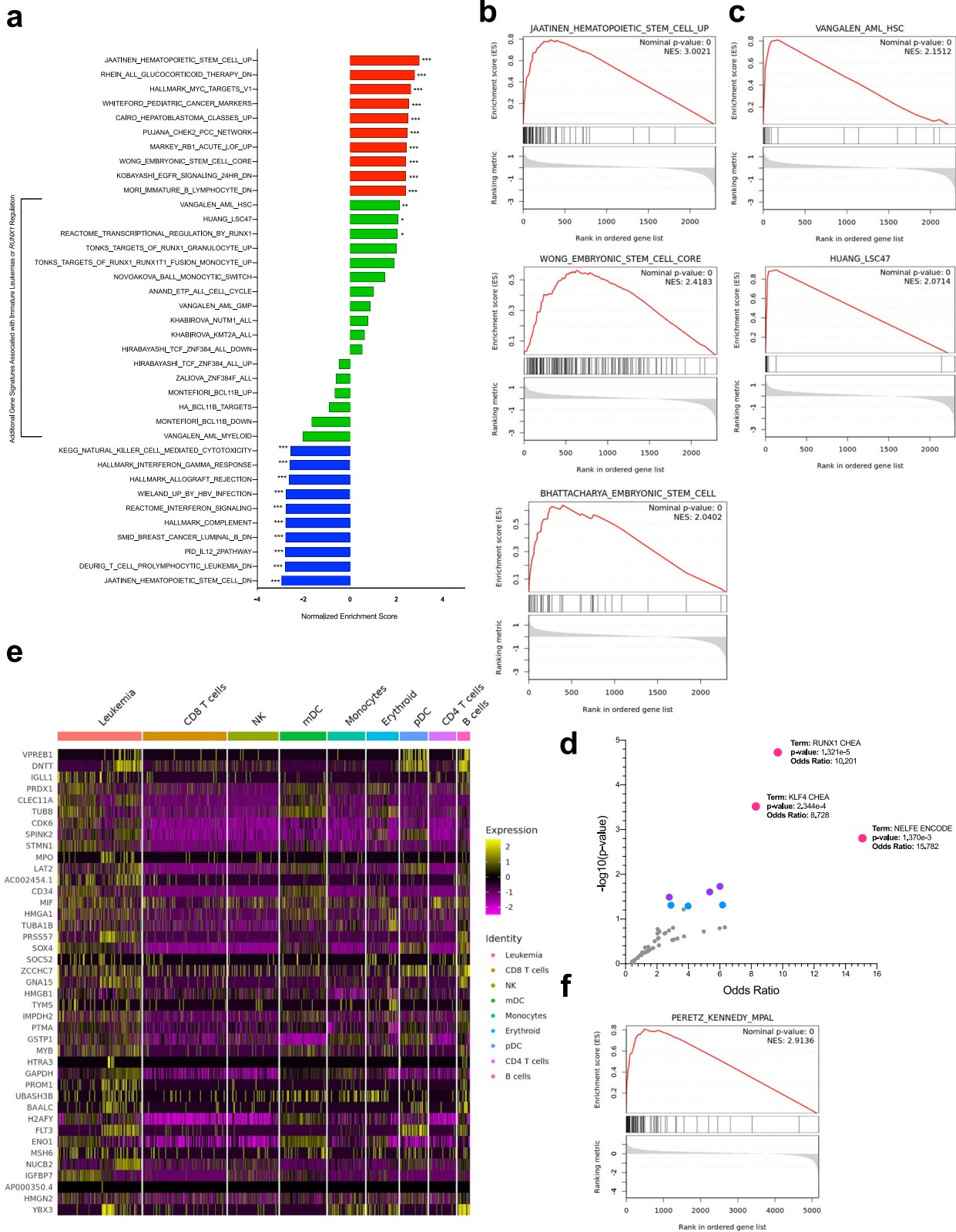

sample from a normal, healthy donor. We identified 10,936 single cells, including 308 HSCs, as identified via scType[35] (Supplementary Fig. 9A). We next re-integrated and clustered the single cells from the normal bone marrow with the 72,131 single cells profiled from our MPAL cohort (Supplementary Fig. 9b, c). The normal-derived HSCs and the leukemic MPAL blast comprised distinct clusters, indicative of distinct transcriptional profiles (Supplementary Fig. 9d). Importantly, the

genes comprising our common MPAL signature, while highly expressed in the MPAL blasts, were not overexpressed in the normal HSCs (Supplementary Fig. 9e). We next performed GSEA to further identify differences in gene expression programs between MPAL blasts and normal HSCs. Relative to normal HSCs, MPAL blasts were significantly enriched for transcriptional programs associated with DNA synthesis and cell cycle regulation, including targets of the DREAM complex[36]

**Fig. 2 | The MPAL transcriptional signature is stem-like, on the continuum of stem-like AML, and reproducible in an independent cohort. a** Barplot of normalized enrichment scores (NES) derived from gene set expression analysis (GSEA) of all single cells in the common leukemia cluster. The top 10 positively enriched gene sets are color-coded in red, the top 10 negatively enriched in blue, and additional gene sets of interest in green. Statistical significance is indicated as \*\*\*q < 0.001, \*\*q < 0.01, \*q < 0.05. Source Data are provided as a Source Data file. **b** Enrichment profile and ranking metric score for three example positively enriched gene sets, all of which are associated with stem cells. Source Data are provided as a Source Data file. **c** Enrichment profile and ranking metric score for the two significant leukemia-specific genes tested, hematopoietic stem cell (HSC)-like AML and leukemia stem cell (LSC)-47. Source Data are provided as a Source Data file. **d** Volcano plot of transcription factors as identified by analysis of the top differentially expressed genes in the common leukemia cluster with the ChIP-x Enrichment Analysis (ChEA) and Encyclopedia of DNA Elements (ENCODE) transcription factor targets databases via enrichr. Points color-coded based on significance as pink: $p < 0.001$, purple: $p < 0.01$, blue: $p < 0.05$. The three most significant gene sets are annotated. $P$ values are two-sided and calculated with Fisher's exact test, where genes are considered independent, and adjusted via the Benjamini–Hochberg method. Source Data are provided as a Source Data file. **e** Heatmap of scaled expression values of top 50 most differentially expressed genes in the common leukemia cluster of our cohort against clustered and annotated single cells from the comparison cohort[28]. Source Data are provided as a Source Data file. **f** Enrichment profile and ranking metric score from GSEA of all single cells in the common leukemia cluster of the comparison cohort. The MPAL gene signature is comprised of the top 50 most differentially expressed genes in the common leukemia cluster of our cohort. The GSEA analysis in (**b**), (**d**), and (**f**) employs a one-sided permutation-based test to determine the significance of gene set enrichment, with raw $p$ values adjusted for multiple testing using the Benjamini–Hochberg procedure to control the false discovery rate (FDR). Source Data are provided as a Source Data file.

and E2F family[37], among others (Supplementary Fig. 9f). Genes comprising our common MPAL signature were also significantly enriched in the MPAL blasts relative to the normal HSCs (NES 2.54, $q = 0.000$), as were the gene signatures derived from HSC-like AML and the LSC-47 (NES 2.37, $q = 0.000$; NES 1.99, $q = 0.002$, respectively) (Supplementary Fig. 9g). Taken together, this confirms that while MPAL blasts are stem-like, they are distinct from non-malignant HSCs and demonstrate aberrant cell cycle regulation.

## MPAL cells demonstrate variable differentiation potential and enhanced proliferation, which predict survival

Given enrichment for genes associated with stemness as well as the lack of enrichment of other known leukemia gene signatures, we sought to apply a more recently developed metric of stemness, CytoTRACE [for cellular (Cyto) Trajectory Reconstruction Analysis using gene Counts and Expression][38], to our SC transcriptional dataset. CytoTRACE is a computational framework for predicting the differentiation potential of a single cell based on transcriptional data about numbers of expressed genes, covariant gene expression, and local neighborhoods of transcriptionally similar cells. CytoTRACE provides a score for each cell representing its stemness within a given dataset, ranging from 0 to 1, with higher scores indicating greater stemness[38]. When applied to our cohort, we found high CytoTRACE scores to be overrepresented in our "leukemia" cluster relative to non-leukemic populations (median CytoTRACE 0.61 vs 0.23 for leukemia vs non-leukemia populations, $p < 2e − 16$) (Fig. 3a).

Across the cohort, CytoTRACE score was moderately correlated with higher CD34 expression, followed by HLA-DR, CD117, and CD33 expression (Spearman correlation coefficient 0.44, 0.25, 0.20, 0.18 for CD34, HLA-DR, CD117, and CD33, respectively) (Fig. 3b, c). For individual patients, the median CytoTRACE score of each patient's leukemia population varied considerably, ranging from 0.13 (least stemlike) to 0.89 (most stemlike). When stratified by median CytoTRACE score of the leukemia population, a higher median CytoTRACE trends toward an inferior OS in our small cohort ($p = 0.053$) (Fig. 3d). Relative to single cells with lower CytoTRACE scores (<0.95), single cells with very high CytoTRACE scores (≥0.95) demonstrated a distinct gene expression profile (Fig. 3e). In a GSEA, cells with CytoTRACE scores ≥0.95 demonstrated upregulation of multiple pathways associated with cellular proliferation, cell cycle dysregulation, and a stem or progenitor-like cell state (Supplementary Fig. 10a). Similarly, pathway enrichment analysis of the conserved genes expressed in the cells with CytoTRACE ≥0.95 against the ChEA and ENCODE databases via enrichr demonstrated significant enrichment for transcription factors in the E2F family, including E2F4 (OR 35.23, $p = 1.38e − 14$), E2F1 (OR 10.58, $p = 6.31e − 6$), and E2F6 (OR = 4.78, $p = 0.0002$) (Supplementary Fig. 10b). The E2F family is involved in DNA synthesis and cell cycle

regulation, with E2F4 being important in embryonic stem cell regulation[39]. Notably, in this analysis NELFE remained enriched (OR 22.5, $p = 7.4e − 6$) and SIN3A, a transcriptional co-repressor implicated in pluripotency[40] and known to regulate MYC activity[41], was also significantly enriched (OR = 8.3, $p = 3.3e − 5$).

## Generation of a CytoTRACE-based prognostic score

We next sought to derive a CytoTRACE-based prognostic metric in patients with MPAL. To generate a CytoTRACE-based score, we compared the differential gene expression of single cells with very high (>= 0.95) vs low (<0.95) CytoTRACE scores. Genes with greatest upregulation in the cells with high CytoTRACE scores were then used to compute a gene set score, which we termed MPAL95. When pseudobulking was applied to all single cells in our cohort, we confirmed that MPAL95 was prognostic for OS (Supplementary Fig. 11A), while the LSC-17, a transcriptionally based risk stratification system previously described in AML[22], was not (Supplementary Fig. 11B). This suggests that, while stem-like AML gene expression is enriched in MPAL blasts, stemness scores defined by other leukemias are not necessarily prognostic in MPAL. Therefore, a MPAL-specific prognostic metric is needed.

## Validation of a CytoTRACE-based score in two independent MPAL patient cohorts

The prognostic ability of MPAL95 was validated using external bulk RNAseq data from two independent patient cohorts: (1) newly diagnosed adult patients with MPAL treated at the First Affiliated Hospital of Soochow University, Suzhou, China, which includes expression profiles for 89 patients with MPAL; 72 patients with available survival data were included in this analysis[42] and (2) newly diagnosed pediatric patients with acute leukemias of ambiguous lineage from the Therapeutically Applicable Research To Generate Effective Treatments (TARGET) initiative, which includes expression profiles for 115 pediatric patients with MPAL; 69 patients with available survival data were included in this analysis[6,43].

Patients from both validation cohorts demonstrated variable MPAL95 scores (Supplementary Fig. 11c, f). In the Soochow University cohort, relative to patients with the lowest MPAL95 scores, patients with high MPAL95 scores demonstrated significantly inferior OS, with a 2-year OS of 44.1% (95% confidence interval 30.3%–58.9%) for patients with high MPAL95 scores vs 70.7% (95% confidence interval 54.0%–98.5%) for patients with low MPAL95 scores ($p = 0.042$; Fig. 3f; Supplementary Fig. 11d). MPAL95 was similarly prognostic in the TARGET cohort, where the 2-year OS was 62.6% (95% CI 50.2%–78.1%) for patients with high MPAL95 scores vs 88.1% (95% CI 73.9%–99.9%) for patients with low MPAL95 scores ($p = 0.018$; Fig. 3g; Supplementary Fig. 11g). Additional clinical variables were available for the TARGET cohort, and the prognostic ability of MPAL95 was preserved in a

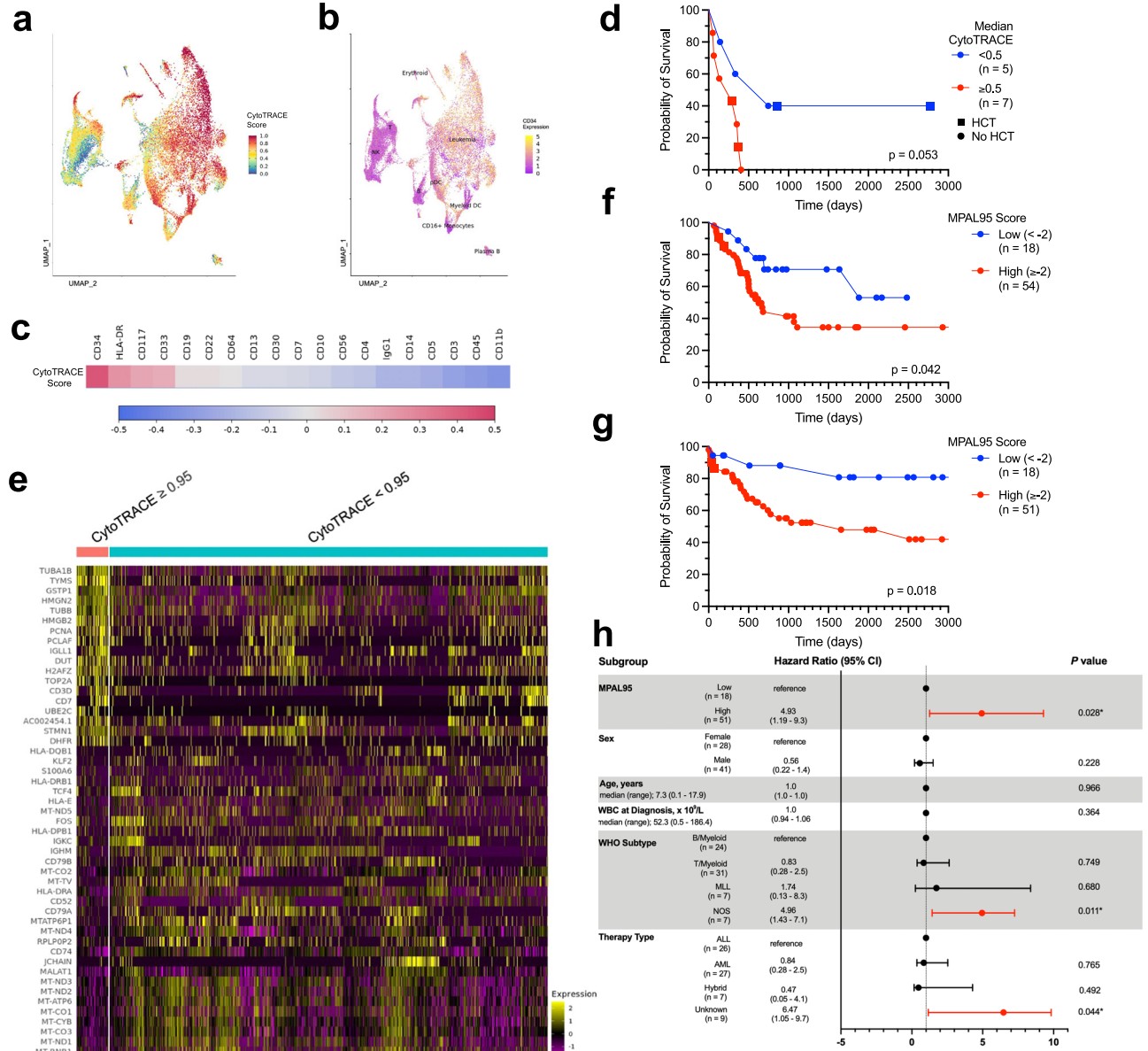

**Fig. 3 | Measures of stemness are prognostic of MPAL patient outcomes. a** RNA-derived UMAP from comprehensive SC CITE-seq analysis of 71,579 cells from 12 patients with MPAL from Fig. 1e. Cells are color-coded based on cytoTRACE score from 0 (most differentiated) to 1 (least differentiated). Source Data are provided as a Source Data file. **b** UMAP from (**a**). Cells are color-coded based on cell-surface expression of CD34 protein. Source Data are provided as a Source Data file. **c** Spearman correlation matrix of CytoTRACE score and cell-surface protein expression. Correlation coefficient is denoted by color coding. Source Data are provided as a Source Data file. **d** Kaplan–Meier estimates of overall survival stratified by median CytoTRACE score <0.5 vs ≥0.5 for 12 adult patients with MPAL. Curves are compared using log-rank tests. Source Data are provided as a Source Data file. **e** Heatmap of scaled expression values for the genes with greatest upregulation in single cells with high cytoTRACE (≥0.95) (left columns) vs low cyto-TRACE (<0.95) (right columns). Source Data are provided as a Source Data file.

**f** Kaplan–Meier estimates of overall survival stratified by MPAL95, a gene set score derived from single-cell transcriptional data, for 72 adult patients from Soochow University[42]. Curves are compared using log-rank tests. Source Data are provided as a Source Data file. **g** Kaplan–Meier estimates of overall survival stratified by MPAL95, a gene set score derived from single-cell transcriptional data, for 69 pediatric patients with MPAL from the TARGET initiative. Curves are compared using log-rank tests. Source Data are provided as a Source Data file. **h** Multivariate Cox proportional hazards model for 69 pediatric patients with MPAL, with the MPAL95 gene signature included. For each variable, the hazard ratio and 95% confidence interval (CI) are graphically depicted. Hazard ratios and 95% confidence intervals are from Cox proportional hazards analyses and *p* values are two-sided and from Wald tests. Statistical significance is indicated as *$p < 0.05$. Source Data are provided as a Source Data file.

multivariable Cox regression model. High MPAL95 score was significantly associated with inferior OS independent of patient age, sex, white blood cell count at diagnosis, WHO subtype, and type of front-line treatment, with a hazard ratio of 4.93 (95% confidence interval 1.19 to 9.3, $p = 0.028$) (Fig. 3h). By contrast, the LSC-17 was not prognostic for OS in either validation cohort (Supplementary Fig. 11e, h). Of note,

consistent with being a pediatric MPAL cohort, the TARGET cohort included genetic subgroups characteristic of MPAL (17.4% *ZNF384*-rearranged, 10.1% *KMT2A*-rearranged, 2.9% *BCL11B*-rearranged) and diverse pathogenic mutation profiles, suggesting that a differentiation-potential prognostic metric may be applicable across genetic subtypes (Supplementary Data 9).

## The CytoTRACE-based score is not prognostic in AML

To assess the specificity of MPAL95 to MPAL vs other leukemias, we next applied MPAL95 to The Cancer Genome Atlas (TCGA) AML cohort ($n = 173$ patients with survival data available)[44] and the BEAT AML cohort ($n = 451$ patients)[45] (Supplementary Fig. 12a, b). Unlike the two MPAL validation cohorts described above, MPAL95 was not prognostic for survival in either AML cohort (Supplementary Fig. 12c, d). As AML blasts can span a spectrum of differentiation states, we also assessed whether MPAL95 was prognostic in the subset of AML patients with immature phenotypes, including HSC-like AML or progenitor-like AML. Interestingly, patients with the lowest MPAL95 scores, representing cells with the least differentiation potential, were not represented in the HSC-like AML subgroup from either the TCGA or BEAT AML cohorts (Supplementary Fig. 12e, g). For the subset of patients with HSC-like AML in the BEAT AML cohort, MPAL95 was prognostic for patients with lower vs higher scores (Supplementary Fig. 12g). By contrast, MPAL95 was not prognostic in the subgroup of progenitor-like AML for either cohort or HSC-like AML in the TCGA cohort (Supplementary Fig. 12e, f, h). Taken together, this suggests that CytoTRACE-based prognostic metrics are preferentially predictive in MPAL but may also have some prognostic ability in other immature leukemias as well.

## The genetic landscape of MPAL

We next turned to evaluate the genetic landscape of our MPAL cohort using DAb-seq. For DAb-seq, we used a panel covering hotspots in 20 genes frequently mutated in leukemia combined with 25 antibody–oligonucleotide conjugates (AOCs) for cell-surface immunophenotypic proteins on hematopoietic cells (Supplementary Data 10, 11)[10–12]. A total of 58,807 individual cells from 14 patients were genotyped, with a median of 4221 cells/sample (range 1093–7245 cells/sample) (Supplementary Data 2).

The mutational landscape for all patients and clones is depicted in Fig. 4a, b. Across the cohort, we identified 27 pathogenic or likely pathogenic mutations within 36 genetically distinct clones (median 2.6 clones/patient, range 0–6); there was no difference in the number of clones between B/myeloid and T/myeloid MPAL (2.8 vs 2.3, $p = 0.66$) (Supplementary Data 12). At the clone level, the most commonly mutated genes were *NRAS*, present in 10 clones (28%), *TP53*, present in 8 clones (22%), and *DNMT3A* and *IDH1*, each present in 7 clones (19%). Clone-level mutational co-occurrence analysis demonstrated the strongest positive association between *NRAS/IDH1* (OR 8.91, $p < 0.0001$), *FLT3/ASXL1* (OR 8.58, $p = 0.008$) and *PTPN11/SF3B1* (OR 4.13, $p = 0.002$); *IDH1/IDH2* were negatively associated (OR −0.58, $p = 0.003$) (Fig. 4c). Except for *DNMT3A/ASXL1*, mutations from the same functional class were infrequently co-mutated in the same single cell and clone; notably, no clones demonstrated more than one distinct signaling mutation.

Using SC DNA sequencing, we reconstructed the evolutionary history of each patient using single cell inference of tumor evolution (SCITE), a probabilistic model to infer genetic phylogeny (Supplementary Fig. 13)[46]. Patients demonstrated diverse phylogenetic trees with both linear and branched architectures. Across the cohort, the most common functional class of founding mutations was epigenetic regulators, at 7/18 (38.8%). This finding in our adult cohort contrasts what has been described in pediatric MPAL, in which transcription factors are the most common truncal mutations[6]. The most common functional class of branch mutations was activated signaling mutations, at 10/25 (40%).

## Genotype alone does not determine immunophenotype

Using DAb-seq, we examined the association between immunophenotype and genetic clonal architecture across all patients. Patients with MPAL demonstrated heterogeneous immunophenotypes among both individual patients and MPAL subtypes (Fig. 4d, e; Supplementary

Fig. 14). Unlike transcription and immunophenotype, where we observed minimal cross-cohort associations, we observed broad genotype–immunophenotype associations across our integrated cohort. These included: associations between *JAK2* mutations and CD71 (point-biserial correlation coefficient 0.8; $p < 7.2e − 8$), *NRAS* and CD34 (point-biserial correlation coefficient 0.89; $p = 0.004$), and *IDH2* and CD11b and CD64 (point-biserial correlation coefficients 0.87 and 0.80; $p = 0.002$ and $p = 0.008$, respectively) (Fig. 4f).

Across the integrated cohort, at the clonal level, we observed considerable inter- and intra-patient heterogeneity (Fig. 4g). For instance, our cohort included four *NRAS*-mutated clones. In 3/4, *NRAS*-mutated cells had significantly increased CD34 expression relative to *NRAS*-wildtype (WT) blasts within the same patient (t-statistics 52.3, 20.1, 22.3; $p = 0.0$, $p = 1.7e − 85$, $p = 3e − 99$); however, in one clone there was no difference in CD34 expression between *NRAS*-mutated vs *NRAS*-WT cells (t-statistic 1.2; $p = 0.25$). Increased expression of other immunophenotypic proteins associated with an immature cell state, including CD38, CD33, CD123, and CD117, was also observed among select *NRAS*-mutated populations (Supplementary Fig. 15a). Similarly, select *DNMT3A, IDH1* and *IDH2* mutated populations were associated with increased expression of CD13 and CD11b, both associated with myeloid/monocytic differentiation, but this pattern was not consistent among all clones with these mutations (Supplementary Fig. 15b). Taken together, these findings suggest that, while some genotype–immunophenotype associations are present in MPAL, genotype alone does not direct the definitional mixed MPAL phenotype.

The heterogeneous association between genotype and immunophenotype was also observed for specific gene mutations; notably, the same mutation does not consistently associate with the same immunophenotype across patients. For example, both Patient 7 and Patient 14 harbor an *IDH2* R140Q mutation. In Patient 7, *IDH2*-mutated cells were significantly associated with increased expression of monocytic markers relative to *IDH2*-WT cells (median CD11b expression 4.12 vs 5.54, $p = 9e − 88$; CD64 2.01 vs 2.89, $p = 1.3e − 34$; CD13 3.34 vs 4.75, $p = 2.3e − 58$; CD14 3.38 vs 3.90, $p = 8.8e − 40$) (Supplementary Fig. 16a, b). Although Patient 14 had the same *IDH2* R140Q mutation, *IDH2*-mutated cells in this patient only demonstrated slightly higher expression of CD11b and did not have higher expression of other monocytic markers (median CD11b expression 3.29 vs 3.67, $p = 0.012$; CD64 1.04 vs 1.11, $p = 0.12$; CD13 3.06 vs 3.20, $p = 0.09$; CD14 2.76 vs 2.88, $p = 0.21$) (Supplementary Fig. 16c, d).

## Progressive mutational acquisition is associated with increase in expression of immunophenotypic markers of immaturity

In addition to the association between genotype and immunophenotype, we also assessed the association between mutational phylogenetic progression and immunophenotypic evolution. Of the 14 patients in our cohort, 9 had at least two stepwise mutational acquisitions identified on SC phylogenetic analysis (Supplementary Fig. 13). For these nine patients, we measured how cell-surface immunophenotypic protein expression changed with progressive acquisition of mutations (Fig. 5a).

Across all nine patients, the maximal change in protein expression was greatest for CD38, CD34, CD33, CD123, and CD117, markers associated with immaturity (HSCs, and in some cases common myeloid or granulocyte–monocyte progenitor cells). Therefore, with progressive mutational acquisition, there was increased expression of these five markers of immaturity. Figure 5b depicts the change in expression of these five immunophenotypic proteins for all nine patients. Despite containing diverse mutations, all nine patients demonstrated significant increase in the expression of at least two of these five proteins with mutational acquisition, and in two patients (Patient 8 and Patient 11), expression of all five proteins increased. Furthermore, for patients with three or more stepwise mutational acquisitions, these immaturity markers often increased multiple times.

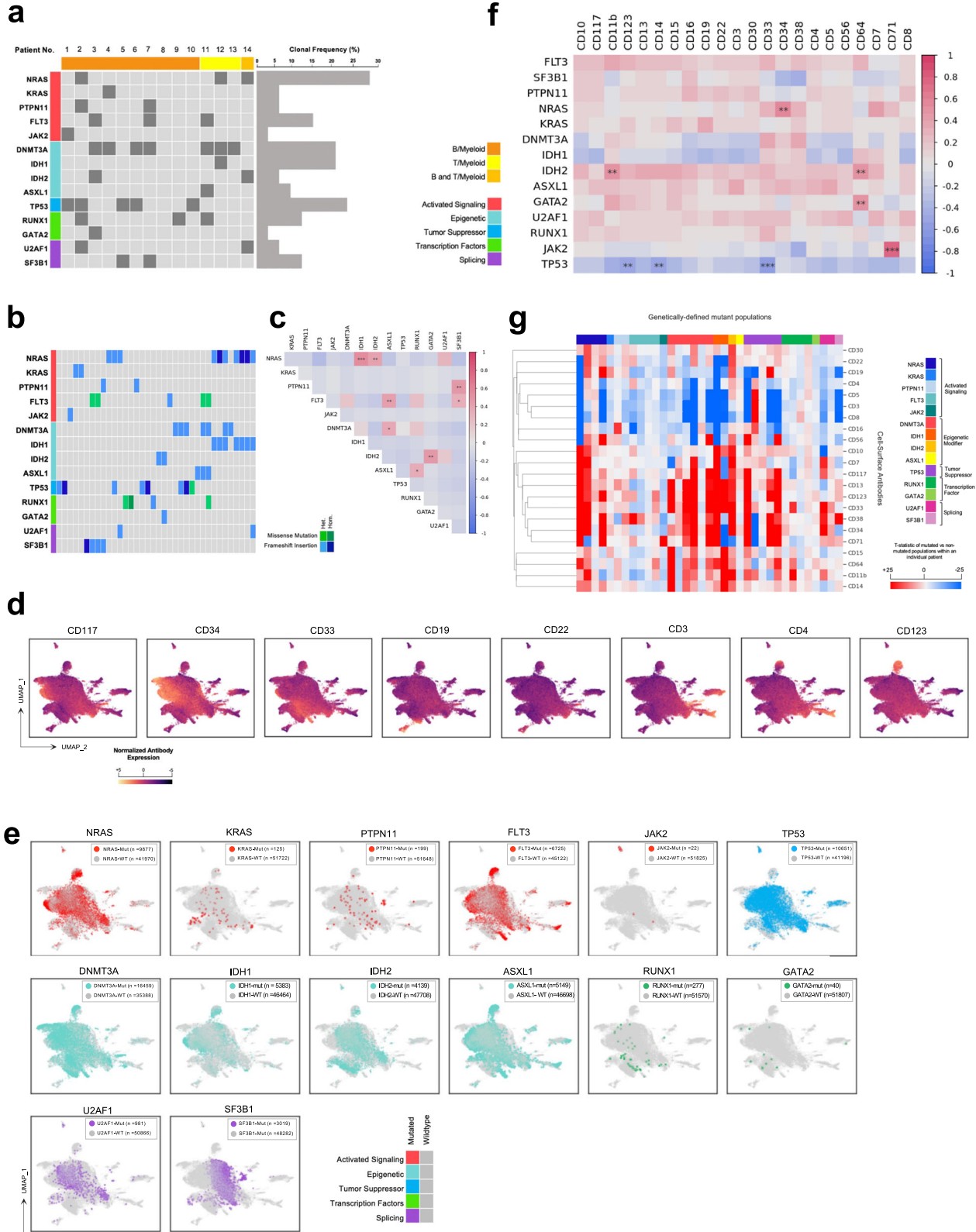

For example, in Patient 8, CD38 and CD34 expression significantly increase with acquisition of a single, heterozygous TP53 mutation, and then significantly increase again with subsequent acquisition of a second, biallelic TP53 mutation. While increased expression of immature markers CD38, CD34, CD33, CD123, and CD117 was the most common immunophenotypic change, evidence of cellular differentiation was seen in select genetic branches. For example, in Patient 12, acquisition of a terminal *DNMT3A* mutation was associated with increased expression of CD11b, CD13, CD14, and CD64, consistent with myeloid and monocytic differentiation (Supplementary Fig. 17). Nonetheless, collectively, these findings suggest that in MPAL leukemic progression, mutational evolution is associated with transition to a more immature immunophenotype and is consistent with the stem-like gene expression profile identified by CITE-seq.

**Fig. 4 | MPAL is comprised of heterogenous genotype–immunophenotype associations. a** Oncoprint of all 14 patients with newly diagnosed MPAL. Each column is a unique patient. Patients (columns) are coded on the top row based on immunophenotypic subtype and mutations (rows) are ordered based on biologic function. Patient-level mutation status is indicated by dark gray (mutated) vs light gray (no detectable mutations) Clonal frequency is based on the total number of clones the mutation was present in, not accounting for zygosity. **b** Oncoprint of 36 genetically defined clones across all 14 patients with MPAL. Each column is a unique clone, and mutations (rows) are color-coded based on the type of mutation and zygosity. Clonal-level mutation status is indicated by heterozygous (Het.) missense (light green), homozygous (Hom.) missense (dark green), Het. frameshift insertion (light blue), Hom. frameshift insertion (dark blue), or no detectable mutations (light gray). **c** Pairwise association of driver mutations identified via SC DNA sequencing across 36 clones in 14 patients with MPAL. For each mutation pair, cooccurrence is summarized as log odds ratio (OR), with positive values indicating cooccurrence and negative values mutual exclusivity. Statistical significance is indicated as *$p < 0.05$; **$p < 0.01$; ***$p < 0.001$. $P$ values are two-sided and calculated using

Fisher's exact test. Source Data are provided as a Source Data file. **d** Immunophenotype-derived UMAP from SC DAb-seq analysis of 51,847 cells from 14 patients. Cells are color-coded based on antibody expression. Selection myeloid and lymphoid markers are shown; all antibodies in the panel are visualized in Supplementary Fig. 14. Source Data are provided as a Source Data file. **e** UMAP from (**d**). Cells are color-coded based on the presence of genetic mutation, with further color coding based on biological function. Source Data are provided as a Source Data file. **f** Spearman correlation matrix across 36 unique genetically defined clones (51,847 single cells) and 22 cell-surface antibodies. Correlation coefficient is denoted by color coding from highly correlated (red) to highly anti-correlated (blue), with significance denoted as *$p < 0.05$; **$p < 0.01$; ***$p < 0.001$. $p$ values are two-sided. Source Data are provided as a Source Data file. **g** Heatmap of $t$-statistics generated by comparing cell-surface antibody expression of mutant vs non-mutant cell populations within an individual patient. To account for differences in expression across patients, comparisons are only made within individual patients, and not across multiple patients. Source Data are provided as a Source Data file.

## Discussion

There is a critical need to improve patient outcomes in MPAL. The historical lack of biologic understanding and subsequent confusion in defining this disease entity remain significant barriers to improving survival. Importantly, there are no consensus guidelines for treatment. In current practice, patients are treated with either ALL- or AML-like chemotherapy, based on empiric assessment rather than knowledge of disease biology[47,48]. A recent analysis suggested matching treatment to ALL- or AML-like chemotherapy based on methylation profiles may improve remission rates[8], but this has not been adopted into clinical practice. Without appropriate definition and comprehensive sub-classification of MPAL, clinical trials to optimize therapy are challenging. Furthermore, no risk stratification for MPAL currently exists. In this context, we use single cell sequencing to dissect the biologic origins of MPAL to provide an improved framework for disease definition and risk stratification.

Although the nomenclature of MPAL suggests that the "mixed phenotype" is the most salient disease component, our data suggest that the mixed immunophenotype of MPAL, while demonstrative of lineage derangement, may have less biologic relevance. Instead, the common stem-like transcriptional signature, and the degree of differentiation potential represented by this signature, likely define MPAL and dictate clinical behavior. Our data suggest MPAL is fundamentally a stem-like leukemia. Our transcriptional analysis highlights enrichment for multiple stem-like signatures, both in our cohort as well as in an independent MPAL cohort characterized by SC RNA sequencing. We also demonstrate upregulation of transcriptional targets of RUNX1 as well as targets of pluripotency factors such as KLF4. *RUNX1* is a key regulator of hematopoiesis[49] and along with recurrent rearrangement/mutation in AML, unmutated *RUNX1*[50] has been implicated in LSC maintenance[51] and leukemogenesis in a variety of AML subtypes[52,53]. In AML, RUNX1 has also been associated with an undifferentiated phenotype (M0)[54] and RUNX1 upregulation has been associated with decreased survival when applied to patients with AML in TCGA[55]. Although RUNX1 is inactivated in some types of acute leukemia, RUNX1 upregulation is implicated in AML1-ETO[52], and in MPAL, RUNX1 signatures have previously been shown to be enriched[28,56]. In this context, our data support a role for RUNX1 activation in driving stem-like gene expression and lineage aberrancy in MPAL. Our pathway enrichment analyses highlighted RUNX1 targets involved in leukemogenesis, including multiple zinc finger proteins (of which ZNF384 is known to be important in MPAL), as well as ALDH[57], ARHGAP[58], ETV[59,60], FANC[61], GATA[62], HOX[63], HSP[64], LMO[65], METTL[66], and TRIM[67] family genes. Finally, we demonstrate enrichment in MPAL for stem-like signatures derived from AML, rather than from ALL, suggesting that MPAL may be more closely related to a stem-like AML[22,68]. This

is particularly relevant clinically, as the current standard of care is to treat MPAL with ALL-like induction therapy[69,70]. Together, our findings situate MPAL among this previous work, as a disease related to a stem-like, therapy-resistant AML.

While there are specific genetic aberrations associated with MPAL[2], our common MPAL gene signature and transcriptional prognostic score is derived from and validated in patients with and without MPAL-associated genetic lesions. Our original cohort of adult patients was genetically heterogenous and included patients with *BCR::ABL1* and *KMT2A* rearrangements, but no patients with *ZNF384* or *BCL11B* rearrangements. Despite this, we identify a unifying gene signature which validates in an independent cohort of adult MPAL patients previously characterized by SC RNAseq[28]. While this independent cohort also lacks MPAL-specific genetic lesions, it is similarly genetically heterogenous and includes patients who received diverse prior treatments[28]. Similarly, we derive a transcriptionally based prognostic metric, MPAL95, that validates in two independent cohorts of adult and pediatric patients with MPAL profiled by bulk RNA sequencing, including patients with *ZNF384*, *BCL11B*, and *KMT2A* rearrangements[42,43]. Notably, MPAL95 was a clear prognostic biomarker for both cohorts. Interestingly, although the adult MPAL cohort found enrichment for an HSC-like signature, this was observed only in patients with *CEBPA* and *NOTCH1* mutations[42]. The fact that MPAL95 validates in two independent cohorts highlights the robustness of this metric despite its derivation from a relatively small cohort. Overall, our findings suggest that our identified stem-like gene signature and associated prognostic score may be broadly applicable across genetic subsets in adult and pediatric patients. Fundamentally, these data highlight the shared stem-like character of MPAL, regardless of genetic subtype.

In myeloid-related acute leukemias, increased stemness has been previously associated with inferior OS[68,71–76]. In MPAL, however, while AML-based stem-like gene signatures are significantly enriched, they are not prognostic. Instead, survival for patients with MPAL is predicted by the gene signature derived from stem-like MPAL blasts with the greatest differentiation potential as defined by CytoTRACE. CytoTRACE is a metric of high differentiation potential in part based on a higher number of expressed genes in the leukemia cell population[38]. In keeping with this definition of stemness and the association of stemness with MPAL phenotypes, MPAL-associated ZNF384 fusion oncoproteins have been found to demonstrate increased promoter occupancy, high chromatin occupancy and high transcriptional activity[4,6]. Similarly, BCL11B-rearranged leukemias are known to display open chromatin profiles enriched for long-term HSPC (LT-HSPC) and activated HSPC (Act-HSPC) signatures[25]. More specifically, in our data, higher differentiation potential as measured by higher CytoTRACE score correlates with a more proliferative,

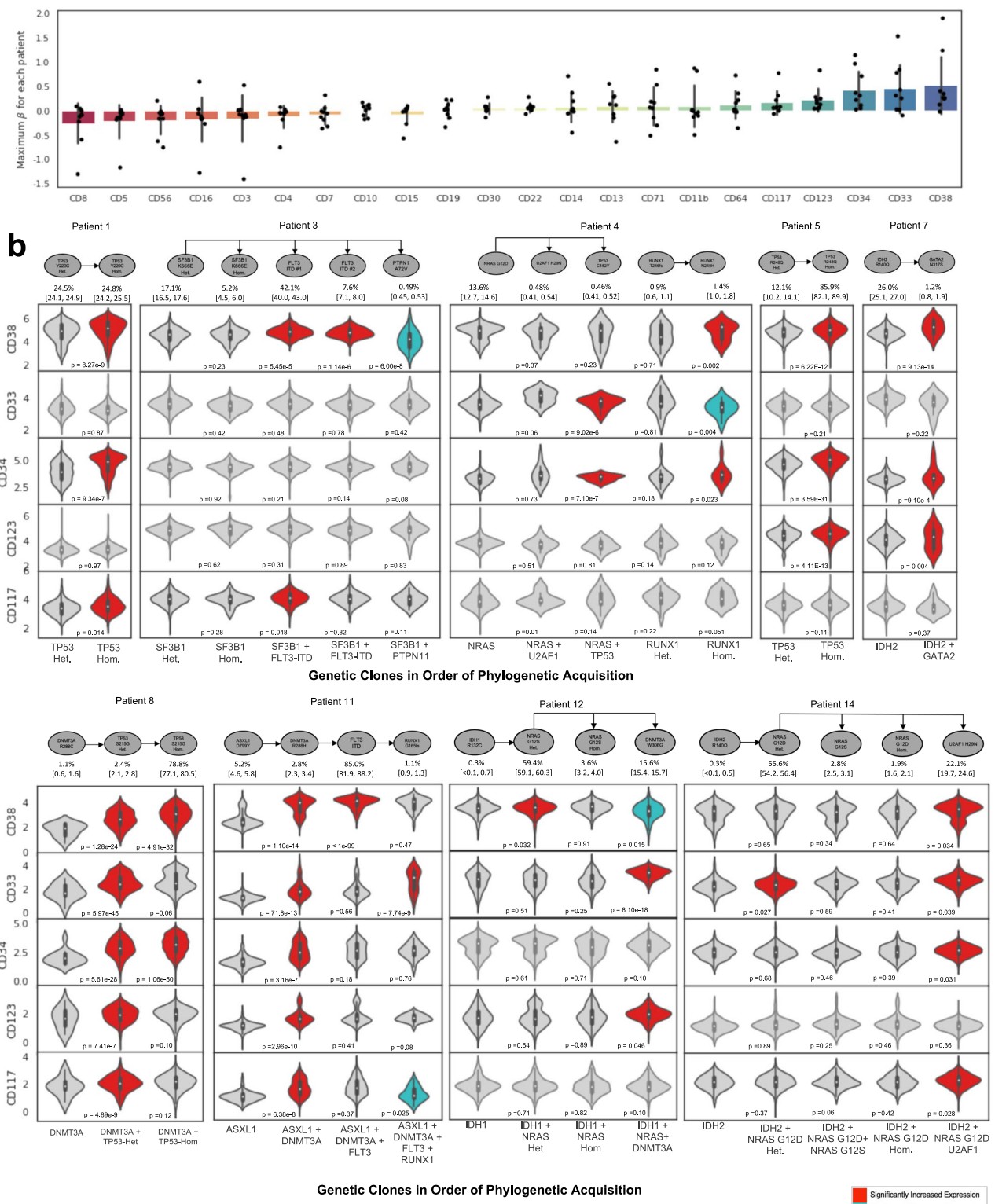

**a**

**b**

Genetic Clones in Order of Phylogenetic Acquisition

Genetic Clones in Order of Phylogenetic Acquisition

aggressive leukemia based on GSEA. This is in opposition to the canonical description of a leukemic stem cell as quiescent[77]. Instead, high CytoTRACE, stem-like, MPAL blasts upregulate transcriptional programs consistent with increased proliferation and cell cycle dysregulation. Furthermore, MPAL cells with high CytoTRACE are enriched for expression of E2F4 transcriptional targets, which have been linked to embryonic stem cell proliferation[39].

In our SC data, the identification of less differentiated and CytoTRACE-high cells was prognostic for patient outcomes. From CytoTRACE-high cells in our SC data, we derived MPAL95, a gene set score applicable to bulk RNAseq data. MPAL95 is not a stemness score; in fact, there is no overlap in genes between MPAL95 and the established AML stemness score LSC-17[68]. Instead, MPAL95 is enriched for the expression of genes associated with proliferation/cell cycle

**Fig. 5 | Association between immunophenotypic evolution and mutational acquisition. a** Barplot with dot-plot overlain depicting maximum *t*-statistic for 22 cell-surface antibodies for each patient across all clones. Bars are defined by the interquartile range, centered at the median, and whiskers indicate 95% confidence interval error bars. For each antibody, antibody expression of all subsequent branch phylogenetic clones are compared to the founding phylogenetic clone, generating a *t*-statistic, and the maximum *t*-statistic for an individual antibody and patient is plotted. Each bar represents one immunophenotypic protein and each overlain dot represents one of nine individual patients. Immunophenotypic proteins are ranked by maximum *t*-statistic across all patients, ranging from CD38 (greatest increase in expression with mutational acquisition across patients) to CD8 (lowest increase in expression). Source Data are provided as a Source Data file. **b** Top: mutation phylogeny of nine patients with MPAL with at least two stepwise

mutational acquisitions identified on single-cell DNA analysis. Each oval represents a genetically distinct subclone and arrows represent cumulative acquisition of mutational events. For each patient, the percentage of each clone among the total number of tumor cells and the 95% credible intervals from the posterior sampling are below each oval. Bottom: violin plots depicting normalized expression of CD38, CD33, CD34, CD123, and CD117 for each subclone represented in the above phylogeny. Violin plots color-coded in red indicate protein expression that has significantly increased with mutational acquisition; plots color-coded in blue indicate a significant decrease in protein expression. Statistical significance is considered *p* < 0.05, with two-sided *p* values calculated using Student's *t*-test and adjusted for multiple comparisons via the Bonferroni method. Het heterozygous, Hom homozygous. All mutations are heterozygous unless specified otherwise. Source Data are provided as a Source Data file.

regulation. Most importantly, MPAL95 is prognostic of survival in our cohort and in two independent MPAL validation cohorts of adult and pediatric patients. Highlighting the specificity of MPAL95, this score is not prognostic in AML datasets—TCGA or BEAT AML, either broadly or in the subset of progenitor-like AML.

Ultimately, the combination of transcriptional and genetic data may provide the most powerful clinically prognostic information. Most leukemias are thought to be driven by a series of successive genetic alterations, culminating in transformation to malignant disease. This canonical road of leukemogenesis, when applied to MPAL, suggests that sequential mutation acquisition leads an MPAL cell to have increased potential for lineage plasticity. Prior investigation into MPAL biology suggested the stem-like nature of MPAL and proposed that mutations in a multipotent progenitor cell led to lineage promiscuity[6]. Interestingly, despite a limited genetic panel, we demonstrate that immunophenotypic markers of immaturity can be gained alongside successive acquisition of mutations in MPAL. In our data, mutational acquisition was associated with increased expression of multiple cell-surface proteins associated with an immature and less differentiated cell state. MPAL may, therefore, arise from a primitive cell, or an MPAL cell may revert to a more primitive phenotype with successive mutational evolution. This suggests that the MPAL cell of origin could span a spectrum of differentiation and supports that a cell's leukemic potential cannot be assigned by immunophenotype alone. Epigenetics may also influence the translation of the genome or transcriptome to lineage marker expression in individual leukemic populations. Regardless, it has been previously shown that mutations do not explain the intra-tumoral heterogeneity of MPAL[6], and our data support this claim.

Multiomic SC analysis allows for direct measurement of cell-surface markers comprising the "mixed" immunophenotype and permits explicit correspondence of immunophenotype with both genetic and transcriptomic profiles. Further, the granularity provided by SC analysis allows for the derivation of a prognostic gene set score applicable to bulk sequencing data. Prior studies have utilized SC analysis both to generate a prognostic metric[78] and to develop cell state scores applicable to bulk RNAseq[79]. Our data similarly demonstrate how specialized SC analysis of even a relatively small patient cohort can lead to broad and clinically relevant conclusions.

As SC DAb-seq and CITE-seq analyses become more common, additional benchtop workflows and/or bioinformatic tools to integrate these diverse data types are warranted. While packages to integrate multiple SC CITE-seq datasets[80] and bulk DNA sequencing with bulk RNA sequencing data[81,82] exist, there remains an unmet need for robust multiomic and tri-omic integration at the single-cell level. This analysis of SC DNA and RNA data was done in parallel and thus, simultaneous linkage of SC DNA and RNA sequencing data for each individual cell is not possible. Our SC analysis was also limited by our targeted genetic panel, and it is possible biologically relevant co-mutational patterns and genotype–immunophenotype associations were not identified.

Nevertheless, this work lays the foundation for a MPAL-specific risk stratification system, which does not currently exist, and supports prospective validation of transcriptionally defined differentiation potential as a prognostic biomarker.

Future clinical studies are needed to validate CytoTRACE and MPAL95 as prognostic tools and to elucidate optimal treatment strategies for MPAL across the span of differentiation potential. Nonetheless, the association of high differentiation potential with poor survival suggests that the potential for lineage plasticity may be advantageous for MPAL cells seeking to evade cytotoxic therapy. Finally, further mechanistic studies will be required to characterize the true cell of origin for MPAL and determine the interplay between genetic, epigenetic, and microenvironmental factors that drive stemness and disease behavior.

## Methods
### Patient samples
Research complies with all relevant ethical regulations. All patients provided written informed consent for sample banking and analysis under protocols approved by the local Institutional Review Boards (either University of California, San Francisco or University of Pennsylvania) and conducted in accordance with the Declaration of Helsinki. Cryopreserved unsorted bone marrow or peripheral blood mononuclear cells from 14 adult patients with newly diagnosed MPAL were included in this study. Patients were diagnosed at either the University of California San Francisco or the University of Pennsylvania from 2006 to 2020, and initial diagnosis was made using WHO criteria operative at the time of diagnosis. All 14 diagnostic samples were analyzed with simultaneous SC DNA and cell-surface protein sequencing; 12 samples were concurrently analyzed with SC RNA and cell-surface protein sequencing (Fig. 1a).

### Single-cell RNA and protein sample preparation, library generation, and sequencing
We performed SC CITE-seq sequencing using a PIPseq platform[9] on 12 diagnostic samples from MPAL patients and one bone marrow sample from a healthy donor (StemCell Technologies). Briefly, cryopreserved cells were thawed, and 1–2 million cells were incubated in 45 µl of Cell Staining Buffer (BioLegend) per million cells with Trustain FcX block (BioLegend) for 15 min on ice. A pool of 19 antibodies (CD3, CD4, CD5, CD7, CD10, CD11b, CD13, CD14, CD19, CD22, CD30, CD33, CD34, CD45, CD56, CD64, CD117, IgG1, HLA-DR) were added (10 µg/mL) and incubated on ice for 60 min (antibody staining performed on MPAL samples only). Cells were resuspended in PBS with 0.04% BSA, combined in a 1:10 ratio with barcoded hydrogel templates (1000 cells/µl), and processed according to PIPseq Single Cell Epitope Sequencing Use Guide Rev 2.0 (FB0002079). Partitioning reagent (Fluent BioSciences) was added to the cell-PIP mixture and vortexed on a custom vortexer (Fluent BioSciences). After the removal of excess partitioning reagent, the emulsion was placed on a dry bath (66 °C for 40 min followed by

4 °C for 11 min) for cell lysis and RNA capture. Emulsions were broken with de-partitioning reagent (Fluent BioSciences), washed, and cDNA synthesis was conducted on the RNA hybridized to PIP templates in bulk. Double-stranded DNA libraries were then enzymatically fragmented and adapters for Illumina sequencing were ligated prior to amplification with appropriate index adapters. The resulting PIPseq libraries were pooled and sequenced using an Illumina NextSeq2000.

## Single-cell CITE-seq data processing and analysis

FASTQ files from single-cell CITE-seq were processed via PIPseeker v0.52 (Fluent), which includes: trimming adapter sequences, demultiplexing data into single cells (BCL Convert, Illumina Basespace dashboard), matching against a list of known barcodes, mapping against the GRCh38.p13 reference transcriptome (Salmon alevin v1.4.0), and separating putative cells from background[9]. Antibody analysis was also processed via PIPseeker v0.52, including error correction, trimming of adapter sequences, mapping to a list of known barcodes, and generating a UMI matrix (CITE-seq Count v1.4.3). Downstream bioinformatics analysis was performed using Seurat 4.3.0 in R. Genes were filtered if detected in <3 cells and cells were filtered based on having low-complexity libraries (feature count <200) or high mitochondrial content (>15%). Unsupervised cell clustering on transcriptional data was performed using Seurat with resolution set to 0.6, and clusters were visualized using the Seurat function *RunUMAP* with default settings. Cell populations were annotated by RNA expression using a combination of scType and clustifyr followed by independent manual confirmation via marker genes[35,83]. Both annotation frameworks agreed on all clusters apart from a population of cells assigned as "cancer cells", "pro-B cells", "progenitor cells", or "unknown" by scType and "CD34+" cells by clustifyr; this cluster was collapsed into a common "leukemia" cluster. Differentially expressed genes for each cluster were determined using Seurat's *FindConservedMarkers, FindAllMarkers*, or *FindMarkers* functions, as appropriate.

## Gene set and pathway enrichment analyses

GSEA were performed using gsea v4.2.3 by comparing single cells annotated as leukemia vs non-leukemia or by comparing single cells within the leukemia cluster with CytoTRACE ≥0.95 vs <0.95; all genes with log2FC threshold ≥0.1 were included[84]. Gene sets used in this study included the molecular signatures database hallmark v2022.1 (50 gene sets) and c2 (6449 gene sets)[13,14] as well as gene sets associated with immature and mature AML and ALL, leukemias undergoing lineage switch, and ZNF384 and BCL11B rearrangements characteristic of MPAL (18 gene sets; Supplementary Data 7). Pathway enrichment analysis was performed using the top 20 greatest upregulated genes by log2FC for both single cells annotated as leukemia vs non-leukemia and for single cells within the leukemia cluster with CytoTRACE ≥0.95 vs <0.95. These gene sets were compared against the ChEA and ENCODE transcription factor targets databases via the enrichr platform[29,30,85,86].

## Comparison with independent single-cell cohort of adult MPAL patients

We used a previously published, independent cohort of SC RNAseq data derived from a cohort of five adult patients with MPAL using the 10x platform[28]. We analyzed the first replicate ("T1") for each of the five patients (MPAL1-5) (GEO Accession Code GSE139369). Downstream bioinformatics processing, including filtering for low-complexity libraries or high mitochondrial content, data integration, unsupervised clustering, cell annotation, and data visualization were performed using Seurat with identical workflow as described above. The top 50 genes upregulated in the common leukemia cluster of our 12-patient cohort were compared against this comparison dataset. We next performed GSEA on the single cells annotated as leukemia vs non-leukemia in the comparison cohort using the 6513 gene sets as described above; we additionally analyzed for enrichment of the top 50 genes upregulated in the common leukemia cluster of our 12-patient cohort ("Peretz_Kennedy_MPAL") (Supplementary Data 8).

## CytoTRACE-based analyses

Differentiation potential was determined using CytoTRACE v0.3.3, with 3000 single cells sub-sampled from the 12 individual patients[38]. To generate MPAL95, a CytoTRACE-derived gene set score, we compared the differential gene expression of single cells with a high CytoTRACE score (≥0.95) vs a low CytoTRACE score (<0.95). Genes with greatest upregulation in the cells with high cytoTRACE scores were used to compute a gene set score, called MPAL95 (Supplementary Data 8), using the first principal component, in an approach similar to that used to compute gene set scores from single-cell transcriptional data in AML[79]. MPAL95 was then applied to bulk RNAseq data from the following: (1) 72 adult patients with MPAL from the recently published Soochow University dataset[42], (2) 69 pediatric patients with MPAL from the TARGET-ALL-P3 dataset; samples were only included if survival outcomes were available[43], (3) 173 adult patients with AML from the TCGA AML cohort[44], and (4) 451 adult patients with AML from the Beat AML cohort[6]. As additional validation, MPAL95 was also applied to pseudo-bulked RNAseq data derived from SC RNAseq data from the 12 adult patients in our cohort. To pseudo-bulk our data, we extracted raw counts from all single cells after quality filtering and then aggregated counts to the sample level.

The TARGET dataset had additional clinical variables available which were included in multivariable survival analysis. These variables included: patient age, sex, white blood cell count at diagnosis, disease classification per WHO classification, and treatment type, classified per TARGET as AML-like, ALL-like, hybrid, or unknown[43]. The TCGA AML and Beat AML datasets were further subsetted to identify patients with HSC-like and progenitor-like AML. To do this, we derived gene set scores from cell state transcriptional signatures[21] as previously described[79]. Patients with the top 10% of each HSC-like and progenitor-like transcriptional scores were included in subset analyses.

## Single-cell DNA and protein sample preparation, library generation, and sequencing

We performed SC DAb-seq using a microfluidic approach with the Tapestri platform (Mission Bio) as previously described[10,87]. Cryopreserved cells were thawed, normalized to 10,000 cells/μL in 180 μL PBS (Corning), and incubated with Human TruStain FcX (BioLegend) and salmon sperm DNA (Invitrogen) for 15 min at 4 °C. A pool of 25 AOCs against 23 antibodies (CD3, CD4, CD5, CD7, CD8, CD10, CD11b, CD13, CD14, CD15, CD16, CD19, CD22, CD30, CD33, CD34, CD38, CD45, CD56, CD64, CD71, CD117, CD123) (Supplementary Data 11) was added (2.5 μg/mL), and cells were incubated for 30 min. Individual samples were also incubated with unique AOCs to provide sample-level identifiers, and groups of 3 samples were pooled together for multiplexed runs. All AOCs were generated as previously described, and successful conjugation was verified using a Bioanalyzer Protein 230 electrophoresis chip (Agilent Technologies, cat. no 5067-1517)[10].

Next, pooled samples were resuspended in cell buffer (Mission Bio), diluted to 4–7e6 cells/mL, and loaded onto a microfluidics cartridge, where individual cells were encapsulated, lysed, and barcoded using the Tapestri instrument. DNA from barcoded cells was amplified via PCR using a targeted panel (Supplementary Data 10). DNA PCR products were isolated, purified with AmpureXP beads (Beckman Coulter), used as a PCR template for library generation, and then repurified with AmpureXP beads. Protein PCR products were isolated via incubation with a 5' Biotin Oligo (IDT), purified using Streptavidin C1 beads (Thermo Fisher Scientific), used as a PCR template for library generation, and then repurified using AmpureXP beads. Both DNA and protein libraries were quantified and quality was assessed via a Qubit

fluorometer (Life Technologies) and Bioanalyzer (Agilent Technologies) prior to sequencing on an Illumina Novaseq.

### Single-cell DAb-seq data processing and analysis

FASTQ files were processed via an open-source pipeline as described previously[10,88]. This analysis pipeline trims adapter sequences, demultiplexes DNA panel amplicons and antibody tags into single cells, and aligns panel reads to the hg19 reference genome. Valid cell barcodes were called using the inflection point of the cell-rank plot in addition to the requirement that 60% of DNA intervals were covered by at least eight reads. Variants were called using GATK (v 4.1.3.0) according to GATK best practices[89]. ITDseek was used to detect *FLT3* internal tandem duplications[90]. For valid cell barcodes, variants were filtered according to quality and sequence depth reported by GATK, with low-quality variants and cells excluded based on the cutoffs of quality score <30, read depth <10, and alternate allele frequency <20%. Cell-surface protein reads were normalized using centered log ratio transformations[91].

### SNP and antibody-based demultiplexing

To de-multiplex individual patients combined into a single sample, we used a custom computational approach incorporating both patient-specific AOC hash antibodies as well as single nucleotide polymorphisms (SNPs) covered by the SC DNA panel[91]. Individual patient samples were stained with unique AOC hash antibodies and then multiplexed into groups of 3. All SNPs were treated as binary (mutated or WT). To identify SNPs that maximally differ between samples, for each multiplexed group, we filtered all SNPs mutated in <10% or >80% of cells. For the remaining SNPs, missing data were imputed based on a majority vote of the binary data from the five nearest neighbors using the kNN function from the VIM package in R. Next, we hierarchically cluster cells using cosine as the distance function and Ward's method for joining clusters and cut the resulting dendogram into three clusters, one for each patient. To refine the SNPs included in clustering, Fisher's exact test was computed between the SNP value and cluster membership across cells; SNPs with $p$ values $< 10^{-12}$ were selected and reclustered in the same hierarchical manner.

Next, SNP-based cell clusters were refined using hash antibody data. Starting with three SNP-based clusters, we add additional clusters by traversing down the hierarchical tree and splitting if there was a significant difference between the current cluster and subsequent split by Hotelling's $T^2$ test with a $p$ value cutoff of $10^{-5}$. Splitting was stopped when there were <10 cells/cluster. Clusters were then assigned to a specific hash antibody by comparing the antibody expression of the cluster to the expected hash background distribution. For each hash antibody, the antibody expression for a multiplexed experiment is expected to be bimodal, with one right mode comprised of antibody-stained cells belonging to a single patient and one left mode comprised of unstained cells. To estimate the expected background antibody distribution, we generated a symmetric distribution by reflecting the data to the left of the left mode about the mode. Clusters were assigned to a specific hash antibody and patient if >50% of cells from that cluster demonstrated hash antibody expression above the 95th percentile of the expected background distribution. A cluster was considered a multiplet if it was assigned to multiple patients. Cells designated as multiplets or unassignable were excluded from downstream analyses.

### Clonal analysis and inference of mutational phylogenies

Following demultiplexing, for individual patients, we analyzed all variants present in >0.1% of cells. Variants were assessed for known or likely pathogenicity via ClinVar and COSMIC databases[24,92] and previously identified, nonintronic somatic variants were included in clonal analyses. Genetic clones were defined as >10 cells possessing identical genotype calls, as per prior SC DNA studies[11,93]. Phylogenetic trees for

individual patients were inferred using SCITE, a probabilistic model for inferring phylogenetic trees[40], using a global false positive rate set to 1% and a platform-provided false-negative rate as per prior SC DNA studies[12]. To define immunophenotypic subpopulations, unsupervised hierarchical clustering was performed using the *Scipy* package in Python, and UMAPs derived from protein expression data were constructed using the *Umap* function with default settings.

In the nine patients that had at least two stepwise mutational acquisitions identified on SC phylogenetic analysis, we measured how cell-surface immunophenotype changed with progressive acquisition of mutations. For each patient, we compared expression of each of the 22 immunophenotypic proteins for the founding genetic clone to all subsequent genetic clones and calculated a $t$-statistic. To identify which cell-surface proteins changed the most with mutational acquisition across the cohort, for each patient, we determined the maximum $t$-statistic for each immunophenotypic protein (Fig. 5a).

### Statistics and reproducibility

Continuous variables were compared using Student's $t$-test or Mann–Whitney $U$ tests and categorical variables were compared using Chi-squared or Fisher's exact tests. To evaluate clone-level cooccurrence, a contingency table was constructed for each mutation pair and the log2-transformed OR computed; Fisher's exact test was used to evaluate statistical significance. The association between individual mutations and cell-surface antibody expression was determined using point-biserial correlations and the association between CytoTRACE and cell-surface antibody expression was determined using Spearman's correlation. Survival analysis was estimated using Kaplan–Meier curves and compared using log-rank tests. Hazard ratios were calculated using the multivariable Cox proportional hazards model. The proportional hazard assumption was tested by examining Schoenfeld residuals using the cox.zph function from the R survival package. All $p$ values for single-cell level comparisons were adjusted via the Bonferroni methods unless otherwise specified. All statistical analyses were performed in R (v. 4.0.2).

### Reporting summary

Further information on research design is available in the Nature Portfolio Reporting Summary linked to this article.

## Data availability

The data as generated here, including raw sequencing data in the form of FASTQ files, have been deposited in NCBI's Gene Expression Omnibus[36] (GEO) and are accessible through GEO series Accession Number GSE232074. Comparison cohorts include single cell RNAseq data from a cohort of five adult patients with MPAL[28] deposited in GEO and accessible through GEO Accession Code GSE139369, bulk RNAseq data from the pediatric database "Therapeutically Applicable Research to Generate Effective Treatments (TARGET)" initiative[43] which is publicly available through [https://ocg.cancer.gov/programs/target/projects/acute-lymphoblastic-leukemia2021.:], and bulk RNAseq data from a cohort of adult patients treated at the First Affiliated Hospital of Soochow University, Suzhou, China, which was requested directly from corresponding authors[42]. Source data are provided with this paper.

## Code availability

Downstream analysis scripts are available at github.com/SmithLabUCSF/MPAL.

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

## Acknowledgements

Sequencing was performed at the UCSF CAT, supported by UCSF PBBR, RRP IMIA, and NIH 1S10OD028511-01 grants. This research was supported in part by the West Charitable Trust. C.A.C.P. is supported (in part) by the National Cancer Institute of the National Institutes of Health

under Award Number K12CA260225. The content is solely the responsibility of the authors and does not necessarily represent the official views of the National Institutes of Health. C.C.S. is the Damon Runyon-Richard Lumsden Foundation Clinical Investigator supported (in part) by the Damon Runyon Cancer Research Foundation (CI-99-18) and is a Leukemia and Lymphoma Society Scholar in Clinical Research.

## Author contributions

C.A.C.P. and V.E.K. contributed equally to this project in its conception and manuscript writing. C.A.C.P. led experimental design and execution and V.E.K. led analysis. A.K., E.T., C.D., Y.X., T.S., and Y.A. ran experiments. A.W., C.L.D., C.E.H., A.A.M.-Z., and R.R. performed or assisted with analysis. Q.W. and H.D. analyzed Chinese patient cohort. I.C.C., K.M.F., and A.A. worked on technology development. A.C.L. and A.E.P. provided samples and assisted with concept development. A.O. mentored and directed bioinformatic and statistical analysis. C.C.S. oversaw all aspects of project conception and completion.

## Competing interests

C.D., Y.X. and K.M.F. are employees of Fluent BioSciences whose technology was used for RNA–protein experiments. C.E.H. is a former employee of Fluent BioSciences. I.C.C. is a shareholder in Fluent BioSciences. A.A. is a co-founder and shareholder of Mission Bio, whose technology was used for DNA–protein experiments, and Fluent BioSciences. All other authors declare no potential conflicts of interest.
