## [Peer Review File · Nature Communications]

Editorial Note: Page 2 and 4 in this Peer Review File have been amended to follow editorial policy for the ECR Co-Review Scheme.

REVIEWER COMMENTS

Reviewer #1 (Remarks to the Author): Expert in single-cell multi-omics, cancer genomics, and computational genomics;

Peretz et al. used DAb-seq and CITE-seq to study Mixed Phenotype Acute Leukemia (MPAL). They found that patients with the highest differentiation potential demonstrated inferior survival. They showed that MPAL blasts expressed a stem-like transcriptional profile, which can be used to predict survival. The area is of interest and the single cell multi-omics data is valuable. However, most of the observations are descriptive, without providing much novel insights. I believe that the present manuscript does not achieve the full potential of what it could provide. I look forward to future additional advancements.

Major comments

1) This study reported the increased stemness of leukemia cells are associated with inferior survival in MPAL. However, this finding is not unique since stemness has been used to predict survival outcomes and response to therapy in various leukemias (van Rhenen et al., 2005 PMID: 16166428; Gentles et al., 2010 PMID: 21177505; Ng et al., 2016 PMID: 27926740; Qin et al., 2021 PMID: 33408321; Ng et al., 2022 PMID: 34872104). It is better to indicate the unique of this study. Does the stemlike genes in this study overlap with the gene list in previous studies such as LSC17 stemness score?

2) Many results in this study were indicated as “Associate Incompletely”, or “Incompletely Associate”, which does not sound like a strong conclusion. Rather than simply show one or two examples to draw conclusions, a much comprehensive and systematic analysis should be conducted to draw conclusions.

3) Since most of the patient samples have been completed scDAb-seq and scCITE-seq, comprehensive integrated analysis of scDAb-seq data and scCITE-seq data from the same patients potentially could provide much biological insight. With the integrated tri-omics data, it is possible to directly compare the effect of mutation on transcriptomics.

4) Detailed information about each sample should be provided. Page14: "Cryopreserved unsorted bone marrow or peripheral blood mononuclear cells" confused me. The tissue source of each data should be specified. E.g., When the samples are collected? before or after treatment? How many weeks after diagnosis? How many weeks after treatment?

5) Page11: Patient 4 was exemplified as “patients with 3 or more stepwise mutational acquisitions”. However, it seems there is no 3 stepwise mutational according to Fig.S11.

Minor comments

- 1) Font size in some figures is too small, like Fig. 1c,g,e;
- 2) Page 4 bottom line: "mSigDB" should be "MSigDB"; also on Page6 line2 of last paragraph.
- 3) Fig3a,c: "CytoTRACE" perhaps should be "CytoTRACE score";
- 4) Page 8 and page 16 says 22 antibodies, but suppl Table 10 lists 25 antibodies (targeting 23 proteins) and 26 barcodes, why these numbers are not consistent with each other.
- 5) Page15, line 3 of para 2: "all genes with $\log_2FC \geq \pm 0.1$ were included" this format may not right, perhaps $abs()$?
- 6) Page16, line 6 of para6, "CD123) (Table S3)" should be "CD123) (Table S10)", as table S3 is for CITE-seq;
- 7) Page16 last line: "Protein PCR products were isolated f via incubation" should remove "f ";
- 8) Page17 line 1: "(ITD)," should be "(IDT),"?
- 9) Page17 last para: "Hotelling’s T2 test" should be "Hotelling’s T-Square test"?
- 10) Fig1a: use "sc DAb-seq" and "sc CITE-seq"?
- 11) Fig 1f,1h, used "adtUMAP_1", instead of "UMAP_1". What’s the difference between UMAP and adtUMAP?
- 12) Add ref or specify the data used, like in Page 20, Fig2E, better add ref[28]?
- 13) Fig4B, lack of color figure legend.
- 14) page21 Fig4C, "*p, .05; **p, .01; ***p, .001" better to be "*p<0.05; **p<0.01; ***p<0.001", so as Fig4D;
- 15) page21 Fig4E, "Comparisons are only made within" what?
- 16) page21 Fig4G,I, "IDH2-wiltype" should be "IDH2-wildtype"

supp:

- 17) Fig.S6b(Suppl P5), "above each bar" should be "blow each bar";
- 18) Fig.S13(Suppl P12), title "Heatmap" should be "Violin plot";

19) Fig.S14G(Suppl P13), "UMAP from S12 E" should be "UMAP from S12 A", as FigS12 has no E panel;

20) Suppl P21, in "Supplementary Figure 6: Gene Set Enrichment Analysis (GSEA)...", "Figure" should be "Table"

21) Table S11, some contents in column 2 is not visible, like "FLT3";

Reviewer #2 (Remarks to the Author): Early-Career Researcher co-reviewer

Reviewer #3 (Remarks to the Author): Expert in leukaemia genomics and single-cell multi-omics

In this paper, Peretz et al characterize at the genetic, molecular and immunophenotypic levels 14 adult mixed phenotype acute leukemia (MPAL) patients samples, using single cell multiomic analysis. They showed that specific MPAL immunophenotypes do not correlate with genetic or molecular profiles. The authors found that MPAL are particularly associated with a stem cell signature and derive a new score (MPAL95) predictive of survival in independent patient cohorts. This score could be used for patients stratification. Overall, this study characterizes at the single cell level a quite large number of patients samples, but some of the findings have a limited novelty and others have to be consolidated.

MAJOR COMMENTS

1. The design of the study which does not implicate any normal hematopoietic stem and progenitor cells raises some issues regarding the specific stem cell signature of MPAL. Comparison of leukemic stem cell signature to normal more mature cells (T cells, B cells, NK, monocytes) will obviously lead to an immature stem cell-like signature. Similarly, higher stemness score is expected when you compare CD34+ cells to other cells. Could they authors compare the leukemic cells signature to normal HSPC to identify specific pathways related to MPAL? Another paper (Wang et al, Am J Hematol, 2023, PMID 36219502) previously identified molecular subgroups by analyzing 176 adult patients with MPAL using NGS and bulk RNA-seq, as well as by performing sc-analysis on 5 patients. They also identified HSC enriched signature but only in some genetic subgroups. This should be discussed, and the paper should be cited.

2. The authors mentioned that a MPAL specific score is needed and derived the MPAL95 score. They show that it performs better than LSC17 and thus conclude that “stemness scores defined by other leukemias are not necessary prognostic in MPAL”. Could the authors apply their score on classical (not MPAL) AML cohort (TCGA/ BeatAML cohorts) to show the specificity of the score? or if it can discriminate between AML with a more immature (HSC like) phenotype and more engaged (progenitor like) phenotype?

3. When looking at the genetic alterations of MPAL, TP53 alterations (either by mutations or deletion of chr17, usually associated with complex karyotype) are highly represented. Did the authors look if it can discriminate the survival of patients? In general, do a high MPAL score is associated with specific genetic or cytogenetic features?

4. RUNX1 seems to be implicated in MPAL with upregulation of RUNX1-regulated transcriptional programs and increased chromatin accessibility of RUNX1 motifs. Usually loss of RUNX1 functions are described in pathogeny of AML. Could the authors identify specific genes regulated by RUNX1 important for leukemogenesis? How RUNX1 is expressed in residual non-leukemic cells compared to leukemic cells in each patient? Is RUNX1 itself highly expressed or specific isoforms?

5. In the clonal architecture inferred through SCITE (supp fig 11, fig 5b), could the authors represent for each clone the number of cells harboring the mutations and the total number of cells analyzed? As well as the probability of each genotype transition?

Also it seems sometimes difficult to conclude with certainty when very few cells are analyzed (exp: IDH2 WT cells in fig4h-i) or studies of GATA2 and JAK2 mutated cells (supp fig 12c). It would help to have a number of cells analyzed below each graph and also to have the number of cells used to do the correlation with the surface markers (figure 4d).

Could the authors also represent for each patient which mutations are present only in immature cells and the ones present also in mature cells (=preleukemic)?

MINOR COMMENTS

1. Another paper analyzing at the sc level a cohort of MPAL patients integrating immunophenotype, transcriptome and ATAC-seq has been published in 2019 (Granja et al, Nature Biotech, 2019). The paper is cited for the implication of RUNX1 but findings of this paper should be better discussed in the context of the current paper.
2. How is determined the leukemia cluster? which signature is used?
3. MPAL95 score to detail in supplementary table
4. Both peripheral blood and bone marrow samples were used and it is known that normal immature cells could have a specific signature depending on their localization. Did the authors look if the samples cluster according to their origin?
5. “proportion of leukemia cluster ... ranged from 4.5% to 10.4%”: it seems quite low as patients are presenting with AML and > 20% blasts. Could the authors provide an explanation?
6. suppl figure 4: CD3 and CD56 do not seem to be expressed by the majority of T- cells and NK cells, respectively. Could the authors discuss this point ?
7. The part on BCL11b could perhaps go into the discussion
8. suppl figure 7: error in the upper GSEA (all patients): the geneset “TONKS_TARGETS_OF_RUNX1_GRANULOCYTE_UP” is indicated 2 times
9. I did not get what are the 10 or 11 populations presented in suppl figure 13, is it one column per patient harboring the mutation?
10. supp fig 14: the panel 14d is the same than panel 14h. There are errors also in the legend (UMAP from S14 and not S12, etc...)

Reviewer #4 (Remarks to the Author): Expert in cancer multi-omics, computational genomics, and immunogenomics

In the study, the authors propose a multi-omic strategy to address the challenges in understanding and treating mixed phenotype acute leukemia (MPAL). They used multi-omic single-cell profiling of 14 adult MPAL patients, revealing that neither genetic profiles nor transcriptomes reliably correlate with specific immunophenotypes of the cancer. However, MPAL patients exhibit a shared stem cell-like transcriptional profile with high differentiation potential. The authors claim to have uncovered a

stem cell-like transcriptional signature in MPAL blasts and to have developed a corresponding score that can be used to predict the survival of MPAL patients. Interestingly, patients with the highest differentiation potential show poorer survival. They replicated this finding in a small independent cohort, suggesting a potential approach for clinical risk stratification.

The article is well-written, especially the introduction, which provides a clear clinical context, helping readers understand the relevance of the information. As far as the authors' summary of the literature is accurate, such results would be valuable to their field, as MPAL seems to lack known biological mechanisms and prognostic tools compared to more common types of acute leukemia. Therefore, a transcriptional metric derived from MPAL blasts that can predict potential patient survival and offer insights into the pathology is of clinical interest. However, to truly assess the veracity of the findings, some modifications and extensions of the analyses presented are necessary before they convincingly support the main conclusions. See the specific comments below.

Major comment

The main weakness of the study is the way in which putative blasts are separated from putative healthy cells in the single-cell data, i.e. the definition of the “leukemia” cluster. The Methods section summarizes this process as follows:

“Cell populations were annotated by RNA expression using a combination of scType and clustifyr followed by independent manual confirmation via marker genes^{63,64}. Both annotation frameworks agreed on all clusters apart from a population of cells assigned as “cancer cells”, “pro-B cells”, “progenitor cells”, or “unknown” by scType and “CD34+” cells by clustifyr; this cluster was collapsed into a common “leukemia” cluster.”

Are there any good reasons to think that the resulting cluster does not contain healthy cells in addition to blasts? This could explain why the cluster appears heterogeneous. If the proportion of healthy cells in this cluster varies between patients, it could also explain the patient-specific trends observed. If these healthy cells are less stem-like than the true leukemia cells, then the proportion of true leukemia cells in the cluster may be a confounding factor in the association between the MPAL95 score and survival: a higher proportion of cancer cells may cause both a higher MPAL95 score and a lower chance of survival.

I have two suggestions to clarify this point:

A) Including data from non-MPAL controls in analyses. If there are no healthy cells in the “leukemia” cluster, then this cluster should be undetectable in single-cell data from healthy subjects. If the association between MPAL95 and survival reflects the specific biology of MPAL, it should be absent from single-cell or bulk data from AML/ALL cases.

B) Looking for further signs of cancer or health in the smaller clusters that were collapsed into the “leukemia” cluster. Maybe some of them can be filtered out for good reasons.

Minor comments

1) Even if the article is well written, the contents are dense and challenging to follow. The structure could be improved by organizing sections and figures around the main ideas and key concepts.

2) Could the authors comment on the size of the cohort (including that of the replication cohort), which seems relatively small? Could this have an impact in the prediction model?

3) Similarly, the raw number of cells from each patient in the “leukemia” cluster is relevant to the interpretation of statistical tests, yet it does not appear anywhere.

4) The number of sequencing reads per patient, and the corresponding ratio of the number of reads to the number of cells, would be useful to interpret clusters, as cells with fewer reads are more likely to be assigned an imprecise or inaccurate cell type.

5) According to the Methods section: “Cryopreserved unsorted bone marrow or peripheral blood mononuclear cells from 14 adult patients with newly diagnosed MPAL were included in this study.” Yet I cannot find any indication of which sample is from bone marrow, and which is from peripheral blood. This seems relevant for the interpretation of results.

6) The Results subsection titled “Generation of a CytoTRACE-based prognostic signature and validation in an independent cohort” mentions: “[...] with a hazard ratio of 4.93 (95% confidence interval 1.19–20.3, $p = 0.028$) (Figure 3G).” Yet the interval reported in the figure itself is “1.19 – 9.3”.

7) There are several mismatches between numbers and plotted confidence intervals in Figure 3G. For instance, several plotted intervals visibly have a negative lower bound, yet the corresponding numbers are positive.

8) The units and ranges of variation of each covariate in Figure 3G should be represented in order to interpret the confidence intervals, as the Cox hazard ratios are factors by which hazard levels change when a covariate changes by one unit.

9) Are there reasons to think that the covariates in Figure 3G satisfy the assumptions of the Cox proportional hazard model?

10) In Figure 4A and Figure 4B, the coloring of tiles is hard to interpret, even with the caption. If the color code of Figure 4B is the one shown in the legend in Figure 4E, then this should be mentioned explicitly in the caption, or shown in the figure itself.

11) The first subsection of Results mentions: “Relative to non-leukemic cells and clusters, the leukemia cluster demonstrated a unique transcriptional signature (Figure 1C, Figure S3, Table S4-5).” In one sense of the term “signature”, this claim is self-evident because the clustering is transcription-based. In another sense, this claim is misleading because the “leukemia” cluster visibly has the weakest signature (strongest heterogeneity).

12) The second subsection of Results mentions: “For many of these subpopulations, the cell type as identified by transcription closely associated with the expected immunophenotype (Figure S4)” This is not visible in Figure S4, because expected associations between cell-surface proteins and cell types are not represented. Also, Seurat “feature plots” may not be the best way of showing this, because dots and clusters overlap. A grid of violin plots (with clusters as columns and cell-surface proteins as rows) would work better.

13) The subsection of Methods titled “CytoTRACE-Based Analyses” mentions: “To pseudo-bulk our data, we first sub-setted the transcriptionally-identified leukemic cell populations, extracted raw counts after quality filtering, and then aggregated counts to the sample level.” Is there a good reason to think that selecting only cells from the “leukemia” cluster makes the pseudo-bulked data more representative of true bulk data? If not, it should include data from all cells.

Reviewer #1

Major comments

1) This study reported the increased stemness of leukemia cells are associated with inferior survival in MPAL. However, this finding is not unique since stemness has been used to predict survival outcomes and response to therapy in various leukemias (van Rhenen et al., 2005 PMID: 16166428; Gentles et al., 2010 PMID: 21177505; Ng et al., 2016 PMID: 27926740; Qin et al., 2021 PMID: 33408321; Ng et al., 2022 PMID: 34872104). It is better to indicate the uniqueness of this study. Does the stemlike genes in this study overlap with the gene list in previous studies such as LSC17 stemness score?

Thank you for helping us to clarify the uniqueness of our findings. We agree with the reviewer's point that stemness has been established, in other leukemias, to correlate with inferior survival, and we have included the additional references provided by the reviewer in our manuscript (*Discussion*, paragraph 4, first sentence).

What we find to be unique about MPAL is that the stem-like MPAL blasts are not all quiescent, as stem cells are generally described in other leukemias. Rather, there is a subset of highly proliferative, yet stem-like cells. To emphasize this important point, we have further clarified our Discussion (paragraph 4):

"More specifically, in our data, higher differentiation potential as measured by higher CytoTRACE score correlates with a more proliferative, aggressive leukemia based on gene set enrichment analysis. This is in opposition to the canonical description of a leukemic stem cell as quiescent. Instead, high CytoTRACE, stem-like, MPAL blasts upregulate transcriptional programs consistent with increased proliferation and cell cycle dysregulation. Furthermore, MPAL cells with high CytoTRACE are enriched for expression of E2F4 transcriptional targets, which have been linked to embryonic stem cell proliferation"

We have derived a score, MPAL95, that can be applied to bulk RNAseq data from MPAL that correlates with MPAL patient survival but is not prognostic in AML. Importantly, though MPAL95 is derived from genes highly expressed in leukemia cells whose overall gene expression is stem-like, MPAL95 is not a stemness score. In fact, there is no overlap between the genes comprising MPAL95 and the AML-derived LSC17 stem cell score. The MPAL95 gene list is instead enriched for genes associated with increased proliferation and cell-cycling. We have updated the discussion to clarify and highlight these findings (*Discussion*, paragraph 5):

"In our SC data, identification of less differentiated and CytoTRACE-high cells was prognostic for patient outcomes. From CytoTRACE-high cells in our SC data, we derived MPAL95, a gene set score applicable to bulk RNAseq data. MPAL95 is not a stemness score; in fact, there is no overlap in genes between MPAL95 and the established AML stemness score LSC17. Instead, MPAL95 is enriched for expression of genes associated with proliferation/cell cycle regulation. Most importantly, MPAL95 is prognostic of survival in our cohort and in two independent MPAL validation cohorts of adult and pediatric

patients. Highlighting the specificity of MPAL95, this score is not prognostic in AML datasets – TCGA or BEAT AML, either broadly or in the subset of progenitor-like AML.”

2) Many results in this study were indicated as “Associate Incompletely”, or “Incompletely Associate”, which does not sound like a strong conclusion. Rather than simply show one or two examples to draw conclusions, a much comprehensive and systematic analysis should be conducted to draw conclusions.

Thank you for helping us emphasize the impact of our findings. We have reorganized our results section to focus on our comprehensive analyses as well as changed our wording to be more definitive about the conclusions we have drawn about our cohort as a whole (*Results*, “Transcription alone does not determine immunophenotype” and *Results*, “Genotype alone does not determine immunophenotype”). We have also emphasized the key conclusions from our integrated analysis: while transcription and genotype may partially associate with immunophenotype within an individual leukemia subpopulation, neither transcription nor genotype alone fully determine the “mixed” phenotype of MPAL in all patients. This conclusion, drawn from our integrated data, has not yet been described in MPAL. In addition to revised text, these revisions also include a new Figure 4D and E, which further emphasize our integrated findings.

While we do feel individual patient examples offer unique insight into MPAL biology, we agree with the reviewer that these examples are of less importance. To that end, we have merged and condensed the sections that previously focused on individual patients. We removed the individual examples from prior Supplementary Figure 16 entirely, have moved all other genotype-immunophenotype single patient examples to the supplementary material, and have de-emphasized transcription-immunophenotype examples accordingly.

3) Since most of the patient samples have been completed scDAb-seq and scCITE-seq, comprehensive integrated analysis of scDAb-seq data and scCITE-seq data from the same patients potentially could provide much biological insight. With the integrated tri-omics data, it is possible to directly compare the effect of mutation on transcriptomics.

We agree with the reviewer that full integration of scDAb-seq and scCITE-seq data, providing true “tri-omic” analysis would greatly enhance our and other SC analyses. To the best of our knowledge, this type of integration has thus far never been done by any group. This is a challenging problem and an unmet need within the larger single cell community. We have used several mathematical models to attempt integration on the MPAL dataset included in this manuscript and have concluded that the overlapping antibodies between the Dab-seq and CITE-seq data do not provide adequate anchors for robust and confident integration.

That said, given the importance of this problem, we are actively piloting several “tri-omic” integration techniques with unpublished SC datasets we have generated in which benchtop sequencing protocols were modified to facilitate potential “tri-omic” integration. These modifications include:

(1) Using larger antibody panels with > 140 ADTS for both DABseq and CITEseq. This will dramatically increase possible integration anchors for each single cell.

(2) “Co-staining” protocols, in which antibodies for Dabseq and CITEseq are applied simultaneously, prior to splitting cell vials for use in both Dabseq and CITEseq. This approach will limit technical batch differences and provide for more similar antibody signals distributions between the 2 technologies.

(3) Directly integrating RNAseq into the Tapestry workflow through additional PCR reagents and modification of PCR workflow. We have piloted this approach using a single RNA fusion transcript, the results of which were presented in at the ASH Annual Meeting in December 2023 (Kennedy et al, Blood 2023; 142 (Supplement 1): 4334. doi: <https://doi.org/10.1182/blood-2023-182454>).

In tandem with these bench-top modifications, we are also pursuing ongoing development of novel statistical frameworks for integrating and clustering Dabseq and CITEseq data, including further development of the established iCluster and iCluster-Bayes platforms with incorporation of zero-inflated and Hurdle models to account for the majority zero data in Dabseq data.

While we feel the above approaches are promising, they represent ongoing and future work. These techniques are regrettably not able to be applied on the MPAL samples in this manuscript as MPAL is a rare disease and the existing samples used in this analysis have now been exhausted. Although true bioinformatic “tri-omic” integration has not been accomplished by our group or any other groups to date, we are optimistic that one of the approaches outlined above will prove successful in future work.

We have further highlighted the importance of this unmet need in our manuscript (*Discussion*, 8th paragraph):

“As SC DAb-seq and CITEseq analyses become more common, additional benchtop workflows and/or bioinformatic tools to integrate these diverse data types are warranted. While packages to integrate multiple SC CITEseq datasets and bulk DNA sequencing with bulk RNA sequencing data exist, there remains an unmet need for robust multiomic and “tri-omic” integration at the single-cell level.”

4) Detailed information about each sample should be provided. Page14: "Cryopreserved unsorted bone marrow or peripheral blood mononuclear cells" confused me. The tissue source of each data should be specified. E.g., When the samples are collected? before or after treatment? How many weeks after diagnosis? How many weeks after treatment?

Thank you for this suggestion, also brought up by the other two reviewers. We have updated Supplementary Table 1 to reflect sample source (bone marrow vs peripheral blood). We also added a new Supplementary Figure 1B to clarify that the common leukemia cluster was composed of cells from both bone marrow and peripheral blood origin and that cells did not cluster based on sample origin. This is referenced in the text:

“Notably, all 12 patients, regardless of MPAL immunophenotypic subtype, contributed to the cluster annotated as leukemia, and the common leukemia cluster contained single cells from diagnostic samples derived from both bone marrow and peripheral blood (Figure S1A-B).” (*Results*, paragraph 2)

All samples in this study were diagnostic and thus were obtained at time of clinical diagnosis. No samples were obtained after treatment. This is emphasized further in the text:

“Cryopreserved unsorted bone marrow or peripheral blood mononuclear cells from 14 adult patients with newly diagnosed MPAL were included in this study” (*Methods*, paragraph 1).

5) Page11: Patient 4 was exemplified as “patients with 3 or more stepwise mutational acquisitions”. However, it seems there is no 3 stepwise mutational according to Fig.S11.

Thank you for pointing out this discrepancy. We have adjusted the text to highlight a different patient, Patient 8, which demonstrated stepwise increase in immature markers CD38 and CD34 with acquisition of both a heterozygous and subsequent homozygous *TP53* mutation.

“For example, in Patient 8, CD38 and CD34 expression significantly increase with acquisition of a single, heterozygous *TP53* mutation, and then significantly increase again with subsequent acquisition of a second, biallelic *TP53* mutation.” (*Results*, final paragraph).

Minor comments

1) Font size in some figures is too small, like Fig.1c,g,e;

This has been corrected; the font size of all heatmaps in Figure 1 has been increased by 25%.

2) Page 4 bottom line: "mSigDB" should be "MSigDB"; also on Page6 line2 of last paragraph.

This has been corrected.

3) Fig3a,c: "CytoTRACE" perhaps should be "CytoTRACE score";

This is a good suggestion; the figure has been changed.

4) Page 8 and page 16 says 22 antibodies, but suppl Table 10 lists 25 antibodies (targeting 23 proteins) and 26 barcodes, why these numbers are not consistent with each other.

Thank you for pointing out this discrepancy, this has been corrected.

5) Page15, line 3 of para 2: "all genes with $\log_2FC \geq \pm 0.1$ were included" this format may not right, perhaps $abs()$?

We have re-written this phrase for clarity: “all genes with \log_2FC threshold ≥ 0.1 were included”.

6) Page16, line 6 of para6, "CD123) (Table S3)" should be "CD123) (Table S10)", as table S3 is for CITE-seq;

Thank you for catching this error; it has been corrected.

7) Page16 last line: "Protein PCR products were isolated f via incubation" should remove "f ";

Thank you for catching this error; it has been corrected.

8) Page17 line 1: "(ITD)," should be "(IDT),"?

Thank you for catching this error; it has been corrected.

9) Page17 last para: "Hotelling's T2 test" should be "Hotelling's T-Square test"?

We have reformatted the 2 as a superscript: Hotelling's T² test

10) Fig1a: use "sc DAb-seq" and "sc CITE-seq"?

This is a good suggestion and we have changed Figure 1A accordingly.

11) Fig 1f,1h, used "adtUMAP_1", instead of "UMAP_1". What's the difference between UMAP and adtUMAP?

This is a good suggestion, and we agree having uniform axes labels will simplify our figures. We have changed the labels of Figure 1F and 1H to read UMAP_1 and UMAP_2.

12) Add ref or specify the data used, like in Page 20, Fig2E, better add ref[28]?

This is an excellent suggestion and we have added in the citation.

13) Fig4B, lack of color figure legend.

Thank you for catching this error; it has been corrected.

14) page21 Fig4C, "*p, .05; **p, .01; *p, .001" better to be "**p<0.05; **p<0.01; ***p<0.001", so as Fig4D**

The figure legend has been changed accordingly.

15) page21 Fig4E, "Comparisons are only made within" what?

Thank you for bringing this to our attention. The figure legend now reads:

“G. Heatmap of T-statistics generated by comparing cell-surface antibody expression of mutant vs non-mutant cell populations within an individual patient. To account for differences in expression across patients, comparisons are only made within individual patients, and not across multiple patients.”

Note this is now Figure 4G as part of our reorganization in order to emphasize our comprehensive, cohort-level analyses and conclusions.

16) page21 Fig4G,I, "IDH2-wiltype" should be "IDH2-wildtype"

Thank you for catching this error; it has been corrected.

17) Fig.S6b(Suppl P5), "above each bar" should be "blow each bar";

Thank you for catching this error; it has been corrected.

18) Fig.S13(Suppl P12), title "Heatmap" should be "Violin plot";

Thank you for catching this error; it has been corrected.

19) Fig.S14G(Suppl P13), "UMAP from S12 E" should be "UMAP from S12 A", as FigS12 has no E panel;

Thank you for catching this error. While we did correct this error, in the process of revising our manuscript to focus on comprehensive and cohort-level analyses and conclusions, we have removed the prior Supplemental Figure 14 entirely.

20) Suppl P21, in "Supplementary Figure 6: Gene Set Enrichment Analysis (GSEA)...", "Figure" should be "Table"

Thank you for catching this error; it has been corrected.

21) Table S11, some contents in column 2 is not visible, like "FLT3"

Thank you for catching this error; it has been corrected. Of note, we believe that this was an error in the automated PDF conversion. We would like to make our supplemental data as easy as possible for other researchers to extract genelist and other data from for any potential secondary analysis, thus we would like to leave supplement as excel sheets. While we realize that the submission process automatically verifies documents with a PDF, please refer to excel for best viewing. Thank you!

Reviewer #3 (Remarks to the Author): Expert in leukaemia genomics and single-cell multi-omics

In this paper, Peretz et al characterize at the genetic, molecular and immunophenotypic levels 14 adult mixed phenotype acute leukemia (MPAL) patients samples, using single cell multiomic analysis. They showed that specific MPAL immunophenotypes do not correlate with genetic or molecular profiles. The authors found that MPAL are particularly associated with a stem cell signature and derive a new score (MPAL95) predictive of survival in independent patient cohorts. This score could be used for patients stratification. Overall, this study characterizes at the single cell level a quite large number of patients samples, but some of the findings have a limited novelty and others have to be consolidated.

MAJOR COMMENTS

1. The design of the study which does not implicate any normal hematopoietic stem and progenitor cells raises some issues regarding the specific stem cell signature of MPAL. Comparison of leukemic stem cell signature to normal more mature cells (T cells, B cells, NK, monocytes) will obviously lead to an immature stem cell-like signature. Similarly, higher stemness score is expected when you compare CD34+ cells to other cells. Could they authors compare the leukemic cells signature to normal HSPC to identify specific pathways related to MPAL?

This is an important point and was raised by Reviewer #4 as well. To compare our MPAL blast population to normal HSCs, we obtained a bone marrow sample from a normal, healthy donor and performed single-cell RNAseq using the same benchtop technology and analysis platform as we used with our MPAL cohort. We obtained a total of 10,936 single cells and identified a subpopulation of normal HSCs. When integrated and re-clustered with our MPAL dataset, the normal HSCs occupied a distinct transcriptional cluster. In addition, the normal HSCs did not upregulate our MPAL leukemic signature. Compared to the normal HSCs, the MPAL leukemia cells were transcriptionally distinct, and demonstrated enrichment for several gene expression signatures associated with cell cycle regulation and proliferation. Taken together, this suggests that, while the MPAL blasts have stem-*like* characteristics, they also demonstrate upregulation of transcriptional pathways distinct from normal, non-malignant stem cells.

We have included these additional analyses in a new Supplementary Figure 9 and in the *Results* section under “The common MPAL gene expression signature is not upregulated in normal hematopoietic stem cells”.

2. Another paper (Wang et al, Am J Hematol, 2023, pmid 36219502) previously identified molecular subgroups by analyzing 176 adult patients with MPAL using NGS and bulk RNA-seq, as well as by performing sc-analysis on 5 patients. They also identified HSC enriched signature but only in some genetic subgroups. This should be discussed, and the paper should be cited.

We agree that Wang et al have generated important data which is highly relevant to our current study. We reached out to the authors and obtained the bulk RNAseq data from the published cohort of adult MPAL patients. When we applied our MPAL95 metric, we found it to be prognostic, providing further validation in addition to our existing pediatric validation cohort. These new results are in our manuscript under *Results*, “Validation of a CytoTRACE-based score in two independent MPAL patient cohorts” as well as in several new panels in Supplementary Figure 11.

We further discussed the findings by Wang et al in our discussion (*Discussion*, paragraph 3):

“Similarly, we derive a transcriptionally based prognostic metric, MPAL95, that validates in two independent cohorts of patients with MPAL profiled by bulk RNA sequencing, including patients with ZNF384, BCL11B, and KMT2A rearrangements. Notably, MPAL95 was a clear prognostic biomarker for both cohorts. Interestingly, although Wang et al. found enrichment for an HSC-like signature in the adult MPAL cohort, this was observed only in patients with CEBPA and NOTCH1 mutations. The fact that MPAL95 validates in two independent cohorts, one adult and one pediatric, highlights the robustness of this metric despite its derivation from a relatively small number of patients characterized by SC sequencing. Overall, our findings suggest that our identified stem-like gene signature and associated prognostic score may be broadly applicable across genetic subsets in adult and pediatric patients. Fundamentally, these data highlight the shared stem-like

character of MPAL, regardless of genetic subtype.”

2. The authors mentioned that a MPAL specific score is needed and derived the MPAL95 score. They show that it performs better than LSC17 and thus conclude that “stemness scores defined by other leukemias are not necessary prognostic in MPAL”. Could the authors apply their score on classical (not MPAL) AML cohort (TCGA/ BeatAML cohorts) to show the specificity of the score? or if it can discriminate between AML with a more immature (HSC like) phenotype and more engaged (progenitor like) phenotype?

These are excellent suggestions. Using the same approach we took to apply MPAL95 to the TARGET MPAL dataset, we also applied MPAL95 to two classical adult AML datasets: the TCGA AML data set (n = 173 patients with both survival data and bulk RNAseq data available) and the BEAT AML data set (n = 451 patients with both survival data and bulk RNAseq data available). We determined MPAL95 was not prognostic for overall survival in either dataset (*Results*, “The CytoTRACE-based score is not prognostic in AML”; also new Supplementary Figure 12.)

The question about the discriminatory ability of MPAL in more immature AML subtypes is very interesting as well. To investigate this, we subsetted both the TCGA and Beat AML datasets by AML differentiation state, using transcriptional signatures published by van Galen et al (PMID 30827681) and a methodology described in detail by Bottomly et al (PMID 35868306). Interestingly, we found that HSC-like AML tended to have higher (less differentiated) MPAL95 scores, and that MPAL95 was prognostic in the HSC-like subset of the Beat AML cohort. MPAL95 was not prognostic in progenitor-like AML from either cohort, or from HSC-like AML from the TCGA cohort. Taken together, we feel these results validate that our MPAL95 prognostic metric is indeed specific to MPAL vs other acute leukemias, but that there may be some degree of continuum between the transcriptional state of MPAL and select stem-like AMLs. The finding further corroborates our prior finding that HSC-like AML and LSC47 gene signatures were significantly enriched in MPAL cells. These conclusions are now summarized in paragraphs 4 and 5 of our Discussion.

3. When looking at the genetic alterations of MPAL, TP53 alterations (either by mutations or deletion of chr17, usually associated with complex karyotype) are highly represented. Did the authors look if it can discriminate the survival of patients? In general, do a high MPAL score is associated with specific genetic or cytogenetic features?

This is a good suggestion. In our small cohort, TP53 mutations were not predictive of overall survival:

It is certainly possible that TP53 would be prognostic in a larger cohort, however,

Mutational data was unfortunately not available or not provided for either of our comparisons (TARGET for our pediatric comparison cohort or First Affiliated Hospital of Soochow University, Suzhou, China for our adult comparison cohort.) We note, however, that only 2 patients from the adult cohort had TP53 alterations (Wang et al).

4. RUNX1 seems to be implicated in MPAL with upregulation of RUNX1-regulated transcriptional programs and increased chromatin accessibility of RUNX1 motifs. Usually loss of RUNX1 functions are described in pathogeny of AML. Could the authors identify specific genes regulated by RUNX1 important for leukemogenesis? How RUNX1 is expressed in residual non-leukemic cells compared to leukemic cells in each patient? Is RUNX1 itself highly expressed or specific isoforms?

Thank you for helping us to highlight our interesting RUNX1 findings. Although loss of RUNX1 function has been described in AML, an enrichment of RUNX1 targets has been previously described in MPAL (Granja et al, 2019 and Merati et al, 2021), which is consistent with our findings. We also determined that *RUNX1* expression is increased in the leukemia cells relative to non-leukemia cells, and have added these results in a new Supplementary Figure 7B.

On the reviewer's suggestion, we reviewed the RUNX1 datasets for specific genes described as being involved in leukemogenesis (including ZNF, ALDH, ARHGAP, ETV, FANC, GATA, HOX, HSP, LMO, METTL, and TRIM family genes), and we have discussed these interesting findings in our manuscript (*Discussion*, paragraph 2):

“We also demonstrate upregulation of transcriptional targets of RUNX1 as well as targets of pluripotency factors such as KLF4. RUNX1 is a key regulator of hematopoiesis and along with recurrent rearrangement/mutation in AML, unmutated RUNX1 has been implicated in LSC maintenance and leukemogenesis in a variety of AML subtypes. In AML, RUNX1 has also been associated with an undifferentiated phenotype (M0) and RUNX1 upregulation has been associated with decreased survival when applied to patients with AML in TCGA. Although RUNX1 is inactivated in some types of acute leukemia, RUNX1 upregulation is implicated in AML1-ETO, and in MPAL, RUNX1 signatures have previously been shown to be enriched. In this context, our data support a

role for RUNX1 activation in driving stem-like gene expression and lineage aberrancy in MPAL. Our pathway enrichment analyses highlighted RUNX1 targets involved in leukemogenesis, including multiple zinc finger proteins (of which ZNF384 is known to be important in MPAL), as well as ALDH, ARHGAP, ETV, FANC, GATA, HOX, HSP, LMO, METTL, and TRIM family genes. ”

5. In the clonal architecture inferred through SCITE (supp fig 11, fig 5b), could the authors represent for each clone the number of cells harboring the mutations and the total number of cells analyzed? As well as the probability of each genotype transition?

This is a good suggestion. We agree adding the total number of cells and the percentage of cells in each subclone are important pieces of data for interpreting the phylogenetic analysis. We have updated Supplemental Figure 13 (previously Supplementary Figure 12) to include this data.

SCITE, the algorithm we used to construct our phylogenies, utilizes a Bayesian Markov-chain Monte Carlo (MCMC) method, and as such, does not generate a single probability estimate for each genotype transition. As an alternative, we added the 95% credible interval from the posterior sampling to illustrate the uncertainty in subclonal size. This metric has been provided in previously published papers using the SCITE algorithm (Morita et al, Nature Communications, 2020).

Also it seems sometimes difficult to conclude with certainty when very few cells are analyzed (exp: IDH2 WT cells in fig4h-i) or studies of GATA2 and JAK2 mutated cells (supp fig 12c). It would help to have a number of cells analyzed below each graph and also to have the number of cells used to do the correlation with the surface markers (figure 4d).

This is again an excellent point. We have updated Figure 4F and Figure S16 (previously Figure 4H-K), Figure 4E (previously Figure S12C), and Figure S16 (previously Figure S14) to include number of analyzed cells for each genotype. The total number of analyzed cells for each patient is indicated in the corresponding figure legend. For the correlation with the surface markers, all analyzed cells (n = 51,847) were included in the analysis. The legend for Figure 4F (previously Figure 4D) has been updated to clarify this point.

Could the authors also represent for each patient which mutations are present only in immature cells and the ones present also in mature cells (=preleukemic)?

At this time, it is not possible to distinguish between preleukemic immature cells vs preleukemic mature cells via cell surface immunophenotype alone. Indeed, in MPAL, many cell surface markers are aberrantly expressed and likely represent a continuum of cell state. While it's certainly possible that some of the mutations assessed are present in preleukemic cells, especially those traditionally associated with CHIP (e.g, DNMT3A, ASXL1, etc), we did restrict our analysis to only known pathogenic mutations thought to occur in leukemic cells.

MINOR COMMENTS

1. Another paper analyzing at the sc level a cohort of MPAL patients integrating immunophenotype, transcriptome and ATAC-seq has been published in 2019 (Granja et al, Nature

Biotech, 2019). The paper is cited for the implication of RUNX1 but findings of this paper should be better discussed in the context of the current paper.

We agree that our findings add to the work of Granja et al, 2019, which looks at 5 patients with MPAL by CITEseq and ATACseq. We used the publicly-available single-cell data from Granja et al to validate our MPAL gene expression signature (*Results*, “The common MPAL gene expression signature is upregulated in an independent cohort”), also detailed in Supplementary Figure 8 and highlighted in the discussion (*Discussion*, paragraph 3):

“...Despite this, we identify a unifying gene signature which validates in an independent cohort of adult MPAL patients previously characterized by SC RNA-seq. While this independent cohort also lacks MPAL-specific genetic lesions, it is similarly genetically heterogenous and includes patients who received diverse prior treatments...”

Our work also builds upon the findings by Granja et al in identifying enrichment for RUNX1 transcriptional programs in MPAL (*Results*, “MPAL cells upregulate RUNX1-regulated gene expression programs”), data also in Supplementary Figure 7. Our results regarding *RUNX1* programming, as well as those by Granja et al, are further discussed (*Discussion*, paragraph 2):

“We also demonstrate upregulation of transcriptional targets of RUNX1 as well as targets of pluripotency factors such as KLF4. RUNX1 is a key regulator of hematopoiesis and along with recurrent rearrangement/mutation in AML, unmutated RUNX1 has been implicated in LSC maintenance⁵¹ and leukemogenesis in a variety of AML subtypes. In AML, RUNX1 has also been associated with an undifferentiated phenotype (M0) and RUNX1 upregulation has been associated with decreased survival when applied to patients with AML in TCGA. Although RUNX1 is inactivated in some types of acute leukemia, RUNX1 upregulation is implicated in AML1-ETO, and in MPAL, RUNX1 signatures have previously been shown to be enriched. In this context, our data support a role for RUNX1 activation in driving stem-like gene expression and lineage aberrancy in MPAL. Our pathway enrichment analyses highlighted RUNX1 targets involved in leukemogenesis, including multiple zinc finger proteins, of which ZNF384 is known to be important in MPAL, as well as ALDH, ARHGAP, ETV, FANC, GATA, HOX, HSP, LMO, METTL, and TRIM family genes.”

2. How is determined the leukemia cluster? which signature is used?

From a technical standpoint, our common MPAL leukemia cluster was identified using 2 annotation frameworks, with our specific methodology detailed in *Methods*, “Single-Cell CITE-seq Data Processing and Analysis”:

“Cell populations were annotated by RNA expression using a combination of scType and clustifyr followed by independent manual confirmation via marker genes. Both annotation frameworks agreed on all clusters apart from a population of cells assigned as “cancer cells”, “pro-B cells”, “progenitor cells”, or “unknown” by scType and “CD34+” cells by clustifyr; this cluster was collapsed into a common “leukemia” cluster.”

Using these annotation rules, we identified a common leukemia cluster as well as a novel MPAL transcriptional signature upregulated in this leukemia cluster, which is now detailed in the new Supplementary Table S8. We determined that our MPAL transcriptional signature was also highly

enriched in an independent cohort of adult MPAL patients (*Results*, “The common MPAL gene expression signature is upregulated in an independent cohort”; also Figure S8) but was not enriched in non-malignant stem cells derived from normal bone marrow (*Results*, “The common MPAL gene expression signature is not upregulated in hematopoietic stem cells”; also Figure S9). The latter analysis, in which we determined that our MPAL signature was not upregulated in normal HSCs, was based on the recommendations of Reviewers #1 and 2, Point #1 and Reviewer #4, Point #1.

3. MPAL95 score to detail in supplementary table

This is an excellent suggestion. We have added Supplementary Table S8 to include this gene set score.

4. Both peripheral blood and bone marrow samples were used and it is known that normal immature cells could have a specific signature depending on their localization. Did the authors look if the samples cluster according to their origin?

This is a good point and was also raised by the other 2 reviewers in various capacities. While normal monocytes and myeloid dendritic cells were more likely to be derived from peripheral blood and normal lymphocytes were more likely to be derived from bone marrow, the common leukemia cluster contained cells from both bone marrow and peripheral blood and did not cluster by origin (bone marrow vs peripheral blood). We have updated Supplementary Table 1 to reflect sample source. We also added a new Supplementary Figure 1B to clarify that the common leukemia cluster was composed of cells from both bone marrow and peripheral blood origin and that cells did not cluster based on sample origin. This is referenced in the text and is also visualized in Supplemental Figure 1B.

“Notably, all 12 patients, regardless of MPAL immunophenotypic subtype, contributed to the cluster annotated as leukemia, and the common leukemia cluster contained single cells from diagnostic samples derived from both bone marrow and peripheral blood (Figure S1A-B).” (*Results*, paragraph 2)

5. “proportion of leukemia cluster ... ranged from 4.5% to 10.4%”: it seems quite low as patients are presenting with AML and > 20% blasts. Could the authors provide an explanation?

Thanks for pointing out this lack of clarity. The percentage referenced is the percentage that each patient contributed to the leukemia cluster (rather than percentage blasts in the diagnostic sample from each patient). This sentence in the methods was re-written for clarity (*Results*, “The transcriptional landscape of MPAL”):

“Each of the 12 patients contributed 4.5%-10.4% (median 8.8%) of the cells in the common leukemia cluster after normalization for number of cells isolated per patient (Table S2)”

6. suppl figure 4: CD3 and CD56 do not seem to be expressed by the majority of T- cells and NK cells, respectively. Could the authors discuss this point?

Thank you for pointing this out. As also mentioned by Reviewer #4 in Point #12, UMAPs are not the best way to visualize the association between cell-surface proteins and cell type, in part due

to the very large number of cells as well as the differences in normalized protein expression distribution across antibodies. Per Reviewer #4's suggestion, we have updated Figure S4 to be a grid of violin plots in which cell-type clusters are columns and cell-surface proteins are rows. With this new visualization, it becomes cleared that CD3 expression is elevated in T cells and CD56 in NK cells, respectively.

7. The part on BCL11b could perhaps go into the discussion

Thank you for this suggestion. However, given that BCL11B rearrangements are associated MPAL and have associated gene expression signatures, we feel it is important to compare these signatures to our identified MPAL signature. Therefore, we have chosen to keep this data in its current location in the manuscript (Results: "MPAL cells upregulate stemlike pathways and are distinct from genetically-defined MPAL subsets").

8. suppl figure 7: error in the upper GSEA (all patients): the geneset "TONKS_TARGETS_OF_RUNX1_Granulocyte_UP" is indicated 2 times

Thank you for catching this error; the panels were inadvertently ordered incorrectly. The figure has been adjusted and is now correct.

9. I did not get what are the 10 or 11 populations presented in suppl figure 13, is it one column per patient harboring the mutation?

Thank you for helping us clarify this figure (now Supplementary Figure 15). Each column represents a population of cells within an individual patient with shared genetic features, i.e. a uniquely mutated population. Thus, one patient may have multiple columns if that patient contains multiple genetically unique subpopulations. In Panel A, only populations of cells with signaling mutations are shown (NRAS, KRAS, PTP11, or FLT3). In Panel B, only populations with epigenetic modifier mutations (DNMT3A, IDH1, IDH2, ASXL1) are shown. We have updated the legend for Figure S15 to further clarify this point:

"Supplementary Figure 15. Array of violin plots comparing distributions of antibody expression for mutant vs wildtype cells.

A. Comparison of distributions of antibody expression for mutant vs wildtype cells across 11 population with signaling mutations (NRAS, KRAS, PTPN11, or FLT3). **Each column represents a unique mutated population within an individual patient.** Each row represents expression of 5 cell surface antibodies with the greatest median T-statistic across all 11 populations (CD34, CD38, CD33, CD123, CD117). The grey half of the split-violin plot represents non-mutated cells and the colorful half of the plot represent mutated cells within an individual patient.

B. Comparison of distributions of antibody expression for mutant vs wildtype cells across 10 population with epigenetic modifier mutations (DNMT3A, IDH1, IDH2, ASXL1). **Each column represents a unique mutated population within an individual patient.** Each row represents the expression of 5 cell surface antibodies with the greatest median T-statistic across all 11 populations (CD33, CD13, CD123, CD11b, CD34). The gray half of the split-violin plot represents non-mutated cells and the colorful half of the plot represent mutated cells within an individual patient. All p-values were adjusted via the Bonferroni method for multiple comparisons."

10. supp fig 14: the panel 14d is the same than panel 14h. There are errors also in the legend (UMAP from S14 and not S12, etc...)

Thank you for catching this error, also pointed out by Reviewer 1, Point #19. While we did correct this error, based on the feedback from Reviewers #1/2, we revised our manuscript to focus on comprehensive and cohort-level analyses and conclusions. In doing this, we de-emphasized our patient-level analyses and thus removed the prior Supplemental Figure 14 entirely.

Reviewer #4 (Remarks to the Author): Expert in cancer multi-omics, computational genomics, and immunogenomics

In the study, the authors propose a multi-omic strategy to address the challenges in understanding and treating mixed phenotype acute leukemia (MPAL). They used multi-omic single-cell profiling of 14 adult MPAL patients, revealing that neither genetic profiles nor transcriptomes reliably correlate with specific immunophenotypes of the cancer. However, MPAL patients exhibit a shared stem cell-like transcriptional profile with high differentiation potential. The authors claim to have uncovered a stem cell-like transcriptional signature in MPAL blasts and to have developed a corresponding score that can be used to predict the survival of MPAL patients. Interestingly, patients with the highest differentiation potential show poorer survival. They replicated this finding in a small independent cohort, suggesting a potential approach for clinical risk stratification.

The article is well-written, especially the introduction, which provides a clear clinical context, helping readers understand the relevance of the information. As far as the authors' summary of the literature is accurate, such results would be valuable to their field, as MPAL seems to lack known biological mechanisms and prognostic tools compared to more common types of acute leukemia. Therefore, a transcriptional metric derived from MPAL blasts that can predict potential patient survival and offer insights into the pathology is of clinical interest. However, to truly assess the veracity of the findings, some modifications and extensions of the analyses presented are necessary before they convincingly support the main conclusions. See the specific comments below.

Major comment

The main weakness of the study is the way in which putative blasts are separated from putative healthy cells in the single-cell data, i.e. the definition of the “leukemia” cluster. The Methods section summarizes this process as follows:

“Cell populations were annotated by RNA expression using a combination of scType and clustifyr followed by independent manual confirmation via marker genes^{63,64}. Both annotation frameworks agreed on all clusters apart from a population of cells assigned as “cancer cells”, “pro-B cells”, “progenitor cells”, or “unknown” by scType and “CD34+” cells by clustifyr; this cluster was collapsed into a common “leukemia” cluster.”

Are there any good reasons to think that the resulting cluster does not contain healthy cells in addition to blasts? This could explain why the cluster appears heterogeneous. If the proportion of

healthy cells in this cluster varies between patients, it could also explain the patient-specific trends observed. If these healthy cells are less stem-like than the true leukemia cells, then the proportion of true leukemia cells in the cluster may be a confounding factor in the association between the MPAL95 score and survival: a higher proportion of cancer cells may cause both a higher MPAL95 score and a lower chance of survival.

I have two suggestions to clarify this point:

A) Including data from non-MPAL controls in analyses. If there are no healthy cells in the “leukemia” cluster, then this cluster should be undetectable in single-cell data from healthy subjects. If the association between MPAL95 and survival reflects the specific biology of MPAL, it should be absent from single-cell or bulk data from AML/ALL cases.

B) Looking for further signs of cancer or health in the smaller clusters that were collapsed into the “leukemia” cluster. Maybe some of them can be filtered out for good reasons.

This is a fair point, and was also touched on by Reviewer #3, Point #1. We particularly appreciate this reviewer’s first suggestion to include data from a non-MPAL control. To implement this suggestion, we obtained a bone marrow sample from a normal, healthy donor and performed single-cell RNAseq using the same benchtop technology and analysis platform as we used with our MPAL patient cohort. We obtained a total of 10,936 single cells and identified a small population of normal hematopoietic stem cells (HSCs). When integrated and re-clustered with our MPAL dataset, the normal HSCs occupied a distinct transcriptional cluster. In addition, the normal HSCs did not upregulate our MPAL leukemic signature, which was enriched in both our cohort as well as the single-cell validation MPAL cohort from Granja et al. Compared to the normal HSCs, the MPAL leukemia cells were transcriptionally distinct, and demonstrated enrichment for several gene expression signatures associated with cell cycle regulation and proliferation, suggesting that although MPAL blasts are stem-*like*, they are transcriptionally distinct from healthy, non-malignant HSCs.

We have included these additional analyses in a new Supplementary Figure 9 and in the *Results* section under “The common MPAL gene expression signature is not upregulated in normal hematopoietic stem cells”.

We also agree that assessing the prognostic ability of MPAL95 in AML data further clarifies the extent to which MPAL95 is specific to MPAL. This point was also raised by Reviewer #3 in Point #2. Using the same approach we took to apply MPAL95 to the TARGET and Soochow University MPAL datasets, we also applied MPAL95 to two classical AML datasets: the TCGA AML data set (n = 173 patients with both survival data and bulk RNAseq data available) and the BEAT AML data set (n = 451 patients with both survival data and bulk RNAseq data available). We determined MPAL95 was not prognostic for overall survival in either dataset. These results are in a new Supplementary Figure 12 and also in the *Results* section under “Validation of a CytoTRACE-based score in two independent MPAL patient cohorts.”

Minor comments

1) Even if the article is well written, the contents are dense and challenging to follow. The structure could be improved by organizing sections and figures around the main ideas and key concepts.

Thank you for commenting that our article is well-written. We agree we have a large amount of rich and detailed data. To better relay our complex findings, we have re-organized our article to focus on cohort-level analyses and themes. These changes include:

- In describing the incomplete association between transcription and immunophenotype, we condensed the results focusing on individual patient examples and merged those results with the cohort-level analyses.
- In describing the incomplete association between genotype and immunophenotype, we similarly condensed the results focusing on individual patient examples, including completely removing the prior Supplemental Figure 14 and moving individual patient data that was previously in Figure 4 to the Supplementary material.

While we do think individual examples are helpful in supporting our conclusions, in condensing these sections, we believe our results become less dense and easier to follow.

2) Could the authors comment on the size of the cohort (including that of the replication cohort), which seems relatively small? Could this have an impact in the prediction model?

While our cohort is small, it is the largest adult MPAL cohort analyzed with single cell sequencing to date. MPAL is a rare disease, and we are pleased to be able to shed additional insight on this understudied pathology. Further, although small numbers certainly have the potential to skew a prediction model, we have validated our MPAL gene expression signature in an independent cohort of adult MPAL patients (Granja et al) and our MPAL prognostic metric, MPAL95, in two different independent and genetically diverse datasets, both pediatric and adult. These results were further emphasized in the discussion.

3) Similarly, the raw number of cells from each patient in the “leukemia” cluster is relevant to the interpretation of statistical tests, yet it does not appear anywhere.

Thank you for this suggestion. We have updated Table S2 to include a new column detailing the number of single cells from each patient contributing to the common leukemia cluster. The table is referenced in manuscript (*Results*, “The transcriptional landscape of MPAL”, paragraph 2):

“Each of the 12 patients contributed 4.5%-10.4% (median 8.8%) of the cells in the common leukemia cluster after normalization for number of cells isolated per patient (Table S2).”

4) The number of sequencing reads per patient, and the corresponding ratio of the number of reads to the number of cells, would be useful to interpret clusters, as cells with fewer reads are more likely to be assigned an imprecise or inaccurate cell type.

Thank you for this suggestion. We have updated Supplementary Table 2 to reflect this information for our CITEseq data, as well as the single-cell RNAseq we obtained from a normal healthy bone marrow.

5) According to the Methods section: “Cryopreserved unsorted bone marrow or peripheral blood mononuclear cells from 14 adult patients with newly diagnosed MPAL were included in this study.” Yet I cannot find any indication of which sample is from bone marrow, and which is from peripheral blood. This seems relevant for the interpretation of results.

This is an important point and was also raised by both of the other reviewers. We have updated Supplementary Table 1 to reflect sample source (bone marrow vs peripheral blood). Per Reviewer 2’s suggestion, we also added a new Supplementary Figure 1B to clarify that the common leukemia cluster was composed of cells from both bone marrow and peripheral blood origin, and that cells did not cluster based on sample origin. This is referenced in the text:

“Notably, all 12 patients, regardless of MPAL immunophenotypic subtype, contributed to the cluster annotated as leukemia, and the common leukemia cluster contained single cells from diagnostic samples derived from both bone marrow and peripheral blood (Figure S1A-B).” (Results, “The transcriptional landscape of MPAL”, paragraph 2).

6) The Results subsection titled “Generation of a CytoTRACE-based prognostic signature and validation in an independent cohort” mentions: “[...] with a hazard ratio of 4.93 (95% confidence interval 1.19–20.3, $p = 0.028$) (Figure 3G).” Yet the interval reported in the figure itself is “1.19 – 9.3”.

Thank you for bringing this to our attention. The numbers shown in Figure 3H (previously Figure 3G) are correct and the manuscript text represents a typographical error. It has been corrected.

7) There are several mismatches between numbers and plotted confidence intervals in Figure 3G. For instance, several plotted intervals visibly have a negative lower bound, yet the corresponding numbers are positive.

Thank you for bringing this to our attention. The numbers shown in Figure 3H (previously Figure 3G) are correct, but it appears the plot itself was incorrectly formatted. This has been corrected.

8) The units and ranges of variation of each covariate in Figure 3G should be represented in order to interpret the confidence intervals, as the Cox hazard ratios are factors by which hazard levels change when a covariate changes by one unit.

This is again a good point. We have added units, medians, and ranges to Figure 3H (previously Figure 3G) for the 2 non-categorical variables, Age and WBC at diagnosis.

9) Are there reasons to think that the covariates in Figure 3G satisfy the assumptions of the Cox proportional hazard model?

Prior to running our Cox regression model, we tested to ensure the covariates satisfied the model assumptions. To do this, we used the Schoenfeld residuals against the transformed time using the `cox.zph` function from the “survival” package in R. The results were as follows, and did not indicate any time-dependent coefficient at an alpha of 0.05:

Covariate	P-value
MPAL95	0.777
Sex	0.827
Age	0.826
WBC.at.Diagnosis	0.098
WHO.subtype	0.087
Therapy.type	0.115
GLOBAL	0.150

We have also updated our manuscript with the following sentence (*Methods*, “Statistics and Reproducibility”):

“The proportional hazard assumption was tested by examining Schoenfeld residuals using the `cox.zph` function from the R survival package”

10) In Figure 4A and Figure 4B, the coloring of tiles is hard to interpret, even with the caption. If the color code of Figure 4B is the one shown in the legend in Figure 4E, then this should be mentioned explicitly in the caption, or shown in the figure itself.

Thank you for this excellent suggestion. We have clarified the color legend in both panels 4A and 4B as well as the figure legend. The color code for mutation classes in Figures 4A, 4B, and Figure 4E are the same; but not Figure 4G (previously Figure 4E).

11) The first subsection of Results mentions: “Relative to non-leukemic cells and clusters, the leukemia cluster demonstrated a unique transcriptional signature (Figure 1C, Figure S3, Table S4-5).” In one sense of the term “signature”, this claim is self-evident because the clustering is transcription-based. In another sense, this claim is misleading because the “leukemia” cluster visibly has the weakest signature (strongest heterogeneity).

We agree with the reviewer that a cluster’s signature is self-evident; however, we find it notable that our signature was also significantly enriched in an independent cohort of adult MPAL patients from Granja et al, suggesting that despite heterogeneity, there is a shared transcriptional program in MPAL. We have clarified language in the results section to clarify that there is a common signature despite heterogeneity in both immunophenotype and genotype:

“Although the nomenclature of MPAL suggests that the ‘mixed phenotype’ is the most salient disease component, our data suggest that the mixed immunophenotype of MPAL, while demonstrative of lineage derangement, may have less biologic relevance. Instead, the common stem-like transcriptional signature, and the degree of differentiation potential represented by this signature, likely define MPAL and dictate clinical behavior.”
(*Discussion*, paragraph 2)

And

“While there are specific genetic aberrations associated with MPAL2, our common MPAL gene signature and transcriptional prognostic score is derived from and validated in patients with and without MPAL-associated genetic lesions. Our original cohort of adult

patients was genetically heterogeneous and included patients with BCR::ABL1 and KMT2A rearrangements, but no patients with ZNF384 or BCL11B rearrangements. Despite this, we identify a unifying gene signature which validates in an independent cohort of adult MPAL patients previously characterized by SC RNA-seq. While this independent cohort also lacks MPAL-specific genetic lesions, it is similarly genetically heterogeneous and includes patients who received diverse prior treatments” (*Discussion*, paragraph 3)

12) The second subsection of Results mentions: “For many of these subpopulations, the cell type as identified by transcription closely associated with the expected immunophenotype (Figure S4)” This is not visible in Figure S4, because expected associations between cell-surface proteins and cell types are not represented. Also, Seurat “feature plots” may not be the best way of showing this, because dots and clusters overlap. A grid of violin plots (with clusters as columns and cell-surface proteins as rows) would work better.

This is an excellent point, also raised by Reviewer #6 in Minor Point #6. We agree that UMAPs generated from Seurat’s featureplot function is not the best way to demonstrate cell-surface protein and cell type association. A grid of violin plots is an excellent visualization suggestion. We have updated Figure S4 accordingly.

13) The subsection of Methods titled “CytoTRACE-Based Analyses” mentions: “To pseudo-bulk our data, we first sub-setted the transcriptionally-identified leukemic cell populations, extracted raw counts after quality filtering, and then aggregated counts to the sample level.” Is there a good reason to think that selecting only cells from the “leukemia” cluster makes the pseudo-bulked data more representative of true bulk data? If not, it should include data from all cells.

We agree with the reviewer. Pseudobulked single-cell RNAseq data is most representative of true bulk RNAseq data when all cell types are included in the analysis, including non-malignant immune cells. We redid our pseudobulk analysis to include all cell types, and have revised the manuscript accordingly (*Results*, “CytoTRACE-Based Analysis”, paragraph 2):

“As additional validation, MPAL95 was also applied to pseudo-bulked RNAseq data derived from SC RNAseq data from the 12 adult patients in our cohort. To pseudo-bulk our data, we extracted raw counts from all single cells after quality filtering and then aggregated counts to the sample level.”

The inclusion of all cell types in our pseudobulk analysis did not significantly change our results. Even with inclusion of additional immune cells, individual patients stratified by MPAL95 the exact same way they did when only the leukemic population was used. In other words, Figure S11A remains unchanged. The inclusion of additional immune cells did slightly alter the stratification of individual patients by LSC17, such that the survival curves in Figure S11B are now changed. Importantly, however, the curves in Figure S11B are not significantly different and LSC17 remains non-prognostic in our pseudobulked MPAL cohort.

REVIEWER COMMENTS

Reviewer #1 (Remarks to the Author):

The authors addressed most of my concerns, however, some of key concerns need the authors to further clarify. Furthermore, the authors' answers to question 2 and 3 may weaken the significance of this research. E.g.

For comment #2, Some results of patient examples were moved to supplementaries, the major conclusions are still not based on a comprehensive and systematic analysis.

For comment #3, the authors indicated sc multi-omics integration has been performed, it is difficult to judge without figures. Some figures/results should be presented.

Reviewer #1 (Remarks on code availability):

It will be much better to write detailed tutorial.

Reviewer #2 (Remarks to the Author):

Reviewer #3 (Remarks to the Author):

The authors have adequately addressed all my comments and concerns.

I have no further remarks on the manuscript, excepted a few minor errors to edit:

- Legend Figure 4 panel E: UMAP from panel D instead of panel F
- Legend supp fig7 panel C: n=9 for MPAL patients without known RUNX1 mutations, instead of n=19
- Legend supp fig8 panel B: ... all single cells in the common leukemia cluster as identified in Figure S8A instead of S5A.
- Legend Supp fig9 panel F: top 10 gene sets instead of top 10 genes
- text line 238: KMT2A-rearranged patients instead of KMTA
- text line 467: “NRAS and CD38”, it should be “NRAS and CD34” according to the figure 4F

Reviewer #4 (Remarks to the Author):

We would like to thank the authors for addressing your comments in this revised manuscript.

Reviewer #4 (Remarks on code availability):

After reviewing the notebooks, we did not identify any significant issues. But we didn't try to install and run the code.

Reviewer #1 (Remarks to the Author):

The authors addressed most of my concerns, however, some of key concerns need the authors to further clarify. Furthermore, the authors' answers to question 2 and 3 may weaken the significance of this research. E.g.

For comment #2, Some results of patient examples were moved to supplementaries, the major conclusions are still not based on a comprehensive and systematic analysis.

We thank the reviewer for their comments and apologize that our former response to this reviewer comment did not sufficiently clarify the comprehensive and systemic nature of our analysis.

To be clear, we are defining “comprehensive” as showing data from all patients in the analyzed cohort. We are defining “systematic” as to have analyzed data from each patient in the same fashion. We have added in additional clarifiers to text and figure legends to highlight this. Whenever possible, we show only integrated data and when this is not possible, we show all the individual data from every patient analyzed in the cohort. We are only using these comprehensive and systematically analyzed data to draw our major conclusions. To be further explicit, for the CITE-Seq analysis, comprehensive analysis of integrated data from all analyzed patients is shown in Figure 1b-e, Figure 2 (all panels), Figure 3a-f, and Supplemental Figures 1-7, 10, 11a-b. For the DAb-seq analysis, comprehensive analysis of integrated data from all analyzed patients is shown in Figure 4 (all panels); Figure 5a (with Figure 5b showing the full data for each individual patient); Supplemental Figures 13 and 17 shows individual data for all analyzed patients meeting the stated criteria. Supplemental Figures 14 and 15 show integrated analysis for the whole cohort. The only figure in the manuscript showing any individual patient level data is Supplemental Figure 16. While we feel this supplemental figure does serve an illustrative purpose to show that in two individual patients, the same *IDH2* mutation results in different immunophenotype, this fact that genotype does not drive immunophenotype is already illustrated in integrated analysis in Figure 4c-g. Given the small size of our cohort, it is not possible to perform further integrated analysis for specific mutations so this is why we chose to use the one patient vignette shown in Supplemental Figure 16. However, this single example is not a major underpinning of our overall conclusion.

For comment #3, the authors indicated sc multi-omics integration has been performed, it is difficult to judge without figures. Some figures/results should be presented.

We apologize for this misunderstanding; SC multiomic integration has not been successfully performed for this data set. The challenges, both technical and bioinformatic, are enumerated in our previous reviewer response. In addition to further clarification regarding the two most significant challenges we encountered and our initial attempts at integration, we have provided several accompanying reviewer-only figures below.

(1) First, the total number of overlapping antibodies provide an insufficient number of integration anchors for robust integration. We have piloted multiple bioinformatic integration strategies. Our conclusion from these initial approaches was that the total number of integration anchors was insufficient for robust integration.

In **Reviewer-only Figure 1** we show our attempt to use scVI (Lopez et al, *Nature Methods*, 2018), a Python library for deep probabilistic analysis of single-cell omics data, which as provided is a CITE-seq/CITE-seq integration framework. To modify for DAbseq, we adjusted the underlying algorithm for binary data, including the algorithm for data missingness.

As seen in **Reviewer-only Figure 1**, which visualizes use of scVI for all cells from all patients in our cohort, modified scVI does not provide adequate integration. While we were able to provide some antibody-based integration (e.g., there is partial overlap in the DAbseq (blue) and CITEseq (orange) cells in panel A), crucially, > 75% of resultant clusters (panel C) were predominately defined by assay (DAbseq versus

CITEseq), not by antibody (as compared to antibody overlays displayed in panel B), confirming that this integration was inadequate for the given data set.

Reviewer-only Figure 1. Attempted DAb-seq and CITEseq multiomic SC integration of all patients in our MPAL cohort using a modified version of scVI (Gayoso et al, Nature Biotechnology, 2022). A. Integration of SC DAbseq (blue) and CITEseq (orange) data based on common antibody integration anchors into a common UMAP framework. B. Overlay of integrated antibodies onto common UMAP framework from panel A. C. Hierarchical clustering of common integrated UMAP. Different clusters are indicated by different colors and numbered as per legend. Clusters do not cluster by antibody but instead predominantly cluster by assay (DAb-seq versus CITE-seq).

In **Reviewer-only Figure 2**, we show our attempt at DAb-seq and CITEseq integration using a modified version of iCluster-Bayes (Mo et al, *Biostatistics*, 2017). iCluster-Bayes provides a simple framework for integrating bulk multi-omic data; to our knowledge, we are the first to attempt modification for single-cell data. To modify iCluster-Bayes, we adjusted RNAseq and DNaseq models to be zero-inflated using Hurdle models, a process previously described in SC CITEseq modeling (Cui et al, *Briefings in Bioinformatics*, 2023).

Reviewer-only Figure 2 visualizes use of iCluster-Bayes for a single example patient in our cohort. In both A and B, vertical black lines separate cells into final clusters. While this model did describe 5 distinct clusters using integrated data, much like our results with using scVI, we determined that 4 out of 5 clusters were predominately defined by assay (DAb-seq versus CITE-seq), not by antibodies. Our conclusion from this approach is that, while iCluster-Bayes represents perhaps the most promising avenue for future integration, sufficient improvement in integration anchors are needed for meaningful clustering.

A**B**
Reviewer-only Figure 2. Attempt at DAb-seq and CITEseq integration for a single example patient using a modified version of iCluster-Bayes (Mo et al, Biostatistics, 2018). **A.** Integration of antibody, RNAseq, and DNaseq from SC DAbseq and CITEseq datasets using iCluster-Bayes as-is. Features of greatest weight in the latent model represent rows and single cells represent columns. **B.** Modification of iCluster-Bayes to account for sparse nature of SC data. Features of greatest weight in the latent model represent rows and single cells represent columns. These modifications significantly reduce the number of salient features, as expected.

(2) **The second challenge we encountered is that the distribution of DAbseq and CITEseq antibodies were distinct.** As shown in **Reviewer-only Figure 3**, the histogram of distribution of un-normalized antibody reads using DAbseq (left) vs CITEseq (right) technologies do not match. Even after applying multiple normalization strategies, including centered log ratio transformation, quantile normalization and

variations (i.e., quintile normalization, partial quantile normalization), TMM normalization, among others, the baseline differences in antibody distribution provide a significant obstacle for robust integration.

Reviewer-only Figure 3. Histogram of distribution of un-normalized CD33 antibody reads using DABseq (left) vs CITEseq (right) technologies.

Given our heightened understanding of the obstacles in obtaining robust multiomic SC integration, we are actively piloting bench-top technologies to solve this problem. In future work, along with bioinformatic integration, we propose alterations to bench-top workflows to improve equivalence of antibody staining and ability to integrate these two technologies, but they *cannot be applied to this dataset*, as these samples have now been exhausted. Bench-top strategies we are piloting in future work include:

- (1) Using larger antibody panels with > 140 ADTS for both DABseq and CITEseq. This will dramatically increase possible integration anchors for each single cell. **We are actively piloting this strategy on a different, non-MPAL, dataset and see this as the most promising avenue for robust integration moving forward.**
- (2) “Co-staining” protocols, in which antibodies for Dabseq and CITEseq are applied simultaneously, prior to splitting cell vials for use in both Dabseq and CITEseq. This approach will limit technical batch differences and provide for more similar antibody signals distributions between the 2 technologies. **We are actively piloting this strategy on a different, non-MPAL, dataset for a separate project.**
- (3) Directly integrating RNAseq into the Tapestry workflow through additional PCR reagents and modification of PCR workflow. **We have already piloted this approach** using a single RNA fusion transcript, the results of which were presented in at the ASH Annual Meeting in December 2023 (Kennedy et al, Blood 2023; 142 (Supplement 1): 4334. doi: <https://doi.org/10.1182/blood-2023-182454>).

In tandem to the above, we are continuing ongoing bioinformatic approaches, including modifications of iCluster-Bayes to provide robust multiomic integration. This will be performed on future datasets using larger antibody panels. As single cell multiomic cross-platform integration has never before (to our knowledge) been performed, once it is successfully performed, an entire manuscript will certainly be devoted to this technique. However, this work is beyond the scope of the current publication.

Reviewer #1 (Remarks on code availability):

It will be much better to write detailed tutorial.

We agree with the reviewer that clear, reproducible code is a cornerstone of academic research. To provide additional clarification to our code, we have further commented the available code on github.

The only novel software used in our analysis is our algorithm for demultiplexing single-cell DNA data using a combination of genomic SNPs and antibodies. This novel demultiplexing algorithm is fully detailed in a separate manuscript, currently in revision at *Bioinformatics*.

The manuscript is available at <https://www.biorxiv.org/content/10.1101/2024.02.07.579345v1> and the accompanying code, including instructions for a detailed vignette and tutorial, are available at <https://www.biorxiv.org/content/10.1101/2024.02.07.579345v1>.

All other packages used in the analysis of this manuscript were published by either other research groups or private companies and have published tutorials and/or vignettes detailing their use. Some examples are below:

- PIPseeker for initial analysis of CITEseq data: <https://www.fluentbio.com/products/pipseeker-software-for-data-analysis/>
- Seurat for downstream analysis of CITEseq data: https://satijalab.org/seurat/articles/get_started.html
- DAbseq pipeline for initial analysis of DAbseq data: <https://github.com/AbateLab/DAb-seq/tree/master>
- SCITE for determining phylogeny from DAbseq data: <https://github.com/cbg-ethz/SCITE/blob/master/README.md>

Reviewer #2 (Remarks to the Author):

Reviewer #3 (Remarks to the Author):

**The authors have adequately addressed all my comments and concerns.
I have no further remarks on the manuscript, excepted a few minor errors to edit:**

- Legend Figure 4 panel E: UMAP from panel D instead of panel F

Thank you for this careful review. This has been corrected.

- Legend supp fig7 panel C: n=9 for MPAL patients without known RUNX1 mutations, instead of n=19

Thank you for this careful review. This has been corrected.

- Legend supp fig8 panel B: ... all single cells in the common leukemia cluster as identified in Figure S8A instead of S5A.

Thank you for this careful review. This has been corrected.

- Legend Supp fig9 panel F: top 10 gene sets instead of top 10 genes

Thank you for this careful review. This has been corrected.

- text line 238: KMT2A-rearranged patients instead of KMTA

Thank you for this careful review. This has been corrected.

- text line 467: “NRAS and CD38”, it should be “NRAS and CD34” according to the figure 4F

Thank you for this careful review. This has been corrected.

Reviewer #4 (Remarks to the Author):

We would like to thank the authors for addressing your comments in this revised manuscript.

Reviewer #4 (Remarks on code availability):

After reviewing the notebooks, we did not identify any significant issues. But we didn't try to install and run the code.

REVIEWERS' COMMENTS

Reviewer #1 (Remarks to the Author):

The authors have addressed all my concerns.

Reviewer #2 (Remarks to the Author):
